# Transform-Triggered Adversarial Examples

**Yaoteng Tan**                                                                ytan082@ucr.edu
*University of California Riverside*

**Zikui Cai**                                                                  zikui@umd.edu
*University of Maryland*

**M. Salman Asif**                                                             sasif@ucr.edu
*University of California Riverside*

**Reviewed on OpenReview:** *https://openreview.net/forum?id=If2OGZMJxs*

## Abstract

Deep neural networks are vulnerable to adversarial attacks, yet most existing attack research focuses on adversarial examples that induce fixed, static mispredictions. In this work, we instead exploit a dynamical adversarial manifold that depends on image transforms, which are a group of functions commonly used for data augmentation, preprocessing, and deployment. We incorporate image transforms into the adversarial optimization process, such that at test-time the same transforms, when applied under malicious conditions, act as triggers that induce diverse adversarial behaviors. We show that a single bounded perturbation can encode behaviors that are selectively activated under different transforms. Our study shows that this transform-dependent property consistently exists across multiple deep network architectures (e.g., CNNs and transformers), computer vision tasks (e.g., image classification and object detection), and a broad range of commonly used image transforms. We further characterize how the number of embeddable targets scales with the transform, the victim architecture, and the perturbation budget. Additionally, to further motivate its real-world relevance, we extend our transform-dependent formulation to a hardware-in-the-loop setting, demonstrating its effectiveness under challenging physical conditions. In summary, we introduce a novel and controllable paradigm for adversarial attack deployment, exposing a previously uncharacterized property in deep neural networks.

## 1 Introduction

Adversarial attacks pose well-known threats to deep networks, and have traditionally been studied through the lens of human-visually imperceptible perturbations that can deceive models into misclassifying inputs (Goodfellow et al., 2015; Kurakin et al., 2016; Wang et al., 2021; Zhu et al., 2024; Li et al., 2024; Guo et al., 2025). In real-world settings, inputs can undergo different transforms due to physical environment variations, such as changes in viewpoint, lighting conditions, and acquisition pipelines, with little to no perceptual change. Prior work has attempted to make adversarial examples robust to such variations by optimizing attack objectives as expectation over a distribution of transforms. For instance, the expectation over transformation (EOT) framework (Athalye et al., 2018) seeks transform-invariant attacks that are robust to various image transforms. Image transforms are also utilized to enhance blackbox transferability of adversarial examples (Zhu et al., 2024; Li et al., 2024; Guo et al., 2025). More recently, TPatch (Zhu et al., 2023) moves beyond passively modeling naturally occurring transformations and studies controllable activation: a physical adversarial patch remains benign under normal imaging, but is activated when an attacker injects an acoustic signal into the inertial sensors of an autonomous vehicle system, inducing a specific motion blur that triggers a single attack outcome. Yet across these existing paradigms: transform-invariant, transform-transferable, or trigger-activated, image transforms are primarily used to preserve, transfer, or ac-

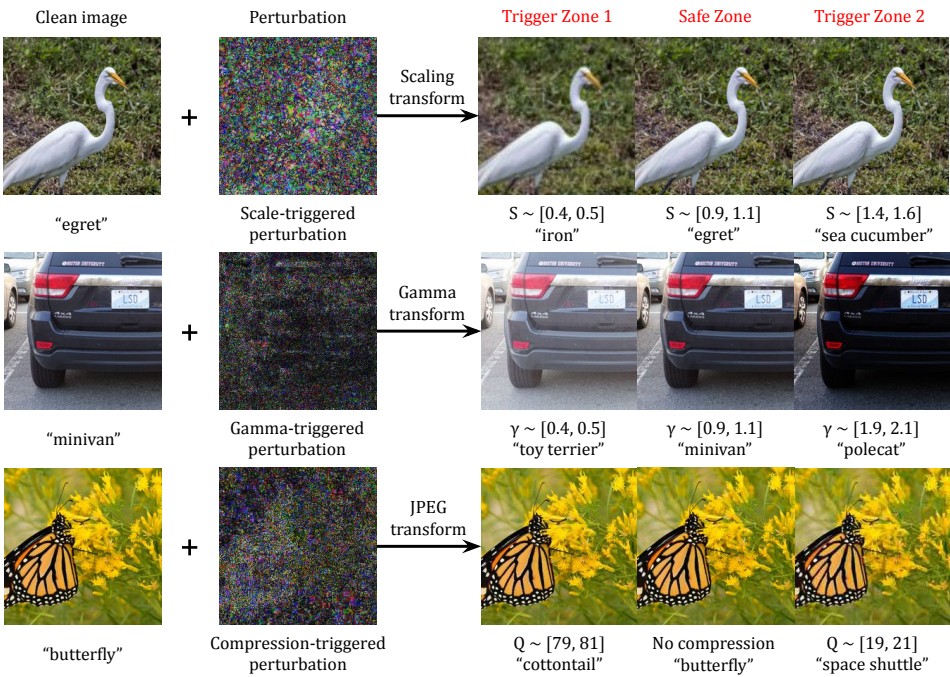

Figure 1: Visual examples of transform-triggered adversarial examples against classifiers. A single adversarial perturbation added to clean image can offer multiple attack effects for the desired presentation conditions. *First row*: Targeted attacks are triggered by scaling around $0.5\times$ and $1.5\times$, with clean label around $1\times$. Scaled images in the first row will have different sizes after scaling, but we present their resized versions for better display. *Second row*: Attacks triggered with $\gamma \sim 0.5 \pm 0.1, 2 \pm 0.1$ in gamma correction, while providing the clean label with $\gamma \sim 1 \pm 0.1$. *Third row*: Attacks triggered with JPEG image compression quality factor $Q \sim 80 \pm 1, 20 \pm 1$, while providing the clean label with no compression. *The perturbation in all examples is bounded by $\ell_\infty \le 8$; the magnitude is amplified $30\times$ for better visualization.*

tivate an attack, rather than to condition distinct malicious outcomes. Consequently, each optimized attack ultimately aims to achieve a single, fixed malicious behavior on the victim model, as summarized in Tab. 1.

Table 1: Comparison of adversarial paradigms in terms of adversarial goals, treatment of transforms, role of transform variations, and induced malicious behaviors. Our adversarial examples condition on transforms, which then play a trigger role to controllably induce multi-malicious behaviors.

| Paradigm | Goal | Transform Treatment | Variation Role | Induced Behavior |
|---|---|---|---|---|
| Standard (Madry et al., 2018) | Deception | Not modeled | – | Single target |
| EOT (Athalye et al., 2018) | Invariance | Marginalized | Nuisance | Single target |
| Transform-transferable | Blackbox transfer | Marginalized | Nuisance | Single target |
| TPatch (Zhu et al., 2023) | Activation | Conditioned | Trigger | Single target |
| Ours | Multi-target control | Conditioned | Trigger | Multi-target |

In this work, we identify a previously uncharacterized adversarially vulnerable manifold that consistently exists in deep networks. Within this manifold, a single post-optimized adversarial example can exhibit multiple adversarial behaviors as it undergoes different transforms at test-time. We aim to systematically characterize this dynamical property of adversarial examples, which we term Transform-Triggered Adversarial Examples. We present visual examples of these adversarial examples in Fig. 1, showcasing how a single perturbation can lead to diverse target labels conditioned on the transforms. In this paradigm, stealthiness is achieved beyond the visually imperceptible perturbation: the perturbation is optimized to keep adversarial effects remain "benign" under normal conditions (e.g., getting the correct label at scales near $1\times$), while

targeted attacks are triggered for a desired set of transform parameters (e.g., scales below or above $0.9-1.1\times$). Unlike existing adversarial paradigms, which seek invariance to transforms or seek invariance to blackbox models, our work reveals a fundamentally different threat model: a single, carefully crafted perturbation can induce a range of targeted attacks, each selectively triggered by transforms that images undergo in traditional imaging and processing pipelines. This transform-triggered behavior offers new attack dimensions beyond additive perturbations; the attack is no longer triggered solely by the presence of an added perturbation (Xiao et al., 2018) or applied transform (Zhu et al., 2023), but instead by the dual control of an additive perturbation and its associated transform.

We demonstrate that transform-triggered adversarial examples widely exist across modern deep network architectures (e.g., CNNs and transformers) for a wide range of image transforms. To further demonstrate the robustness of such attacks in the physical domain, we construct a display–camera hardware pipeline that enables the deployment, capture, and optimization of transform-triggered adversarial examples through a noisy real-world environment. This setup further supports that our transform-triggered attacks are not merely precision artifacts due to input transformations, but are desired malicious behaviors purposely introduced with our proposed formulation.

We summarize our main contributions as follows:

- We introduce transform-triggered adversarial examples, a novel class of adversarial examples that can induce multiple targeted attacks.
- We showcase the versatility and utility of the proposed adversarial examples through extensive experiments across models, computer vision tasks, and transforms. Our experiments demonstrate that a single perturbation can trigger diverse adversarial behaviors in a controllable manner.
- We demonstrate physical attacks with transform-triggered adversarial examples through a camera-in-the-loop pipeline. Our results show that the transform-triggered adversarial examples are not merely artifacts of digital precision, but that they represent a robust feature space that persists even when transforms are physically induced.

Our work characterizes the property of adversarial examples: *a single bounded perturbation can encode multiple distinct target behaviors, each selectively activated by a different transform applied to the input.* This goes beyond the conventional view that an input is either benign or adversarial: the same perturbation can be metamorphic, exhibiting different targeted behaviors as the transform varies, while remaining benign under normal conditions. We show this property is real, consistent across architectures, tasks, and transforms, and practically relevant to deployment pipelines where such "transform triggers" can occur, making it worth documenting and characterizing systematically.

## 2 Related Work

### 2.1 Adversarial attacks and defenses

Adversarial attacks were initially introduced as visually imperceptible perturbations in digital space, capable of causing false predictions in untargeted attacks or inducing desired misclassifications in targeted attacks (Goodfellow et al., 2015; Moosavi-Dezfooli et al., 2017). Adversarial vulnerabilities have been found across various neural architectures (e.g., CNN, transformer), and tasks (e.g., classification, object detection) (Xie et al., 2017; Mahmood et al., 2021). In conventional paradigms, a single optimized perturbation can achieve a (single) fixed attack outcome, offering limited control over adversarial effects. Likewise, existing defenses operate under the assumption that an input is either benign or adversarial, without considering dynamic adversarial behaviors (Raff et al., 2019; Dong et al., 2019). A recent step toward controllable adversarial behavior is TPatch (Zhu et al., 2023), a physical adversarial patch that remains benign under normal conditions but is activated when the attacker injects an acoustic signal into the camera inertial sensors, inducing a specific motion-blur trigger. Zhu et al. demonstrate that conditional, trigger-based activation of adversarial behavior is feasible, but operates with a single trigger and a single target behavior in a specific physical-attack scenario. In contrast, our work aims to characterize a previously uncharacterized structural property of deep networks: that a *single* imperceptible perturbation can encode *multiple* targeted behaviors,

each conditionally activated at the post-optimization stage by ordinary image transforms encountered in standard imaging and pre-processing pipelines, without any attacker-side hardware.

## 2.2 Image transformation in adversarial attacks

In adversarial attack literature, image transforms have primarily been leveraged for two purposes: (i) Generating transform-invariant adversarial examples that remain effective under input corruptions and physical environment variations (EOT) (Athalye et al., 2018; Kurakin et al., 2018; Lennon et al., 2021), as well as transferable adversarial examples that remain effective across models other than the surrogate used during attack generation (Zou et al., 2020; Wang et al., 2021; 2023b; Xie et al., 2019; Dong et al., 2019; Lin et al., 2020). This line of work seeks robust adversarial examples that provide consistent attack effects under various conditions, and is the setting in which the EOT formulation underlying our optimization was originally introduced. (ii) Generating adversarial examples via simple geometric transforms (Pei et al., 2017; Chen et al., 2020; Zhu et al., 2024; Li et al., 2024). This line of work optimizes over transformation parameters, so that a slightly transformed image causes misprediction. While these methods provide new attacks against deep networks through simple image transforms, the attacks are limited to certain transformation parameters, may change the image semantic content (Xiao et al., 2018), or may cause deviation from natural image color distribution (Chen et al., 2020). In contrast, our work exploits the more general and diverse image transformations, covering spatial, photometric, and compression transformations to craft adversarial perturbations capable of dynamically altering their effects based on the transform trigger applied.

# 3 Methodology

We study a class of adversarial examples whose malicious behavior is conditionally activated by image transforms. Unlike conventional adversarial examples that induce a fixed misprediction, the adversarial effect here is *triggered* by specific transform parameters, while remaining benign under others.

## 3.1 Problem statement

Suppose $f$ denotes a trained model, $x \in \mathbb{R}^{H \times W \times 3}$ an input image with ground-truth label $y$, and $\delta$ an additive perturbation constrained within an $\ell_p$-norm ball $\|\delta\|_p \leq \varepsilon$. We denote $\tau(\cdot; \theta)$ as an image transform (e.g., rotation, scaling, or perspective), with $\theta \in \Theta$ as the transform parameters. We further assume (and require) that applying such transforms to clean images should preserve semantic content and output of a properly trained model; that is,

$$f(\tau(x; \theta)) \approx y, \quad \forall \theta \in \Theta. \tag{1}$$

Importantly, equation 1 defines the scope of our work: $\Theta$ is chosen to keep clean transformed images semantically recognizable and to avoid transform-induced collapse in clean accuracy. As summarized in Tab. 12, most selected transforms preserve clean accuracy. Thus, observed attack success is primarily attributable to the desired adversarial effects rather than the transform alone.

**Definition 1** (Transform-Triggered Adversarial Example). Given a model $f$, an input $x$ with label $y$, and a transform $\tau(\cdot; \theta)$, a perturbation $\delta$ is *transform-triggered* if we have two disjoint parameter sets $\Theta_{\text{safe}}$ and $\Theta_{\text{trigger}}$ such that

$$f\big(\tau(x + \delta; \theta)\big) = \begin{cases} y, & \forall \theta \in \Theta_{\text{safe}}, \\ y^{\star}, & \forall \theta \in \Theta_{\text{trigger}}, \ y^{\star} \neq y. \end{cases}$$

This definition formalizes adversarial perturbations whose effect is dormant under normal conditions but activated when specific transform triggers are applied. Our objective is to find $\delta$ that deceives the model into predicting desired target labels $y^{\star}$ under different transform triggers in digital and physical attacks.

## 3.2 Proposed method

**Optimization objective.** To synthesize desired perturbations for digital attacks, we incorporate the transform in the adversarial optimization process. We consider a set of transform parameters $\boldsymbol{\theta} = \{\theta_i\}_{i=1}^{N}$ and

corresponding target labels $\mathbf{y}^\star = \{y_i^\star\}_{i=1}^N$, each $(\theta_i, y_i^\star)$ pair specifies the desired model behavior under transform $\theta_i$. For certain parameters, $y_i^\star$ may coincide with the ground-truth label $y$, enforcing benign behavior.

We seek a single perturbation $\delta$ that satisfies all specified behaviors by solving the following optimization problem:

$$\min_\delta \sum_{i=1}^N \mathcal{L}\big(f(\tau(x + \delta; \theta_i)), y_i^\star\big) \text{ s.t. } \|\delta\|_p \leq \varepsilon, \tag{2}$$

where $\mathcal{L}$ denotes the task loss (e.g., cross-entropy for classification). At test-time, the input examples $x + \delta$, or $\tau(x + \delta; \theta_i)$, $\theta_i \in \Theta_{\text{safe}}$ should appear "benign" (i.e., $f\big(\tau(x + \delta; \theta)\big) = y$). When applying a transform with trigger parameters $\theta_i \in \Theta_{\text{trigger}}$, the corresponding adversarial behaviors will be activated.

**Robustness to parameter variations.** To ensure robustness against small deviations in transform parameters, we adopt an expectation-over-transform (EOT) formulation (Athalye et al., 2018). Instead of optimizing for discrete parameters, we optimize over neighborhoods around each target parameter:

$$\min_\delta \ \sum_{i=1}^N \mathbb{E}_{\theta_i \sim \mathcal{U}_r(\bar{\theta}_i)} \mathcal{L}\big(f(\tau(x + \delta; \theta_i)), y_i^\star\big) \text{ s.t. } \|\delta\|_p \leq \varepsilon, \tag{3}$$

where $\mathcal{U}_r(\bar{\theta}_i)$ denotes a uniform distribution over parameters within radius $r$ of some predefined $\bar{\theta}_i$ (for all $i$). This formulation yields perturbations that remain effective over continuous ranges of transform parameters in digital and physical attacks and improves transferability to blackbox models, consistent with prior observations that transforms enhance adversarial generalization (Xie et al., 2019; Lin et al., 2020; Wang et al., 2021). The full step-by-step optimization procedure for synthesizing transform-triggered adversarial examples in the digital setting is given in algorithm 1.

**Transform functions.** To enable gradient-based optimization, we assume the transform function $\tau(\cdot; \theta)$ is differentiable (or approximately differentiable) with respect to the input. This allows computing gradients via the chain rule: $\nabla_\delta \mathcal{L} = \frac{\partial \mathcal{L}}{\partial f} \frac{\partial f}{\partial \tau} \frac{\partial \tau}{\partial \delta}$. Moreover, transforms are assumed to be deterministic with respect to $\theta$, ensuring that adversarial behavior is desirably induced by the $\theta$ rather than presenting stochastic effects.

### 3.3 Physical attacks with hardware-in-the-loop

To further validate that equation 3 yields robust conditional adversarial features, rather than precision artifacts introduced by transforms, we extend the formulation to a real hardware prototype. We incorporate an explicit imaging forward model $\mathcal{A} : \mathbb{R}^{H \times W \times 3} \to \mathbb{R}^{H \times W \times 3}$ in the optimization problem as

$$\min_\delta \ \sum_{i=1}^N \mathbb{E}_{\theta_i \sim \mathcal{U}_r(\bar{\theta}_i)} \mathcal{L}(f(\mathcal{A}(\tau(x + \delta; \theta_i))), y_i^\star) \text{ s.t. } \|\delta\|_p \leq \varepsilon. \tag{4}$$

Let us represent $\mathbf{p}$ and $\mathbf{s}$ as proxies for $\tau(x + \delta, \theta)$ and $\mathcal{A}(\tau(x + \delta, \theta))$ in equation 4, where $\mathbf{p} \in \mathbb{R}^{H \times W \times 3}$ represents a physical planar scene and $\mathbf{s} \in \mathbb{R}^{H \times W \times 3}$ represents the digital sensor image. We represent mapping from $\mathbf{p}$ to $\mathbf{s}$ as $\mathbf{s} = \mathcal{A}(\mathbf{p})$, where we assume Nyquist sampling in both planes. The optimization problem in equation 4 requires a differentiable model of $\mathcal{A}$. The complete imaging model needs to represent different stages of the imaging process: "scene $\to$ optics $\to$ sensor $\to$ ISP". The true operator $\mathcal{A}$ models the optics, sensor, and ISP stages, but it is challenging to calibrate precisely in practice (mainly because of blackbox nature of ISP and inherent noises in the imaging process). We therefore approximate this pipeline using three parameterized steps, covering variations of **pixel location shifts, color shifts** and **sensing noise**: (i) the geometric map between the scene and sensor planes, represented by a homography matrix $H$; (ii) the camera color response, represented using a color–correction matrix $C \in \mathbb{R}^{3 \times 3}$; and (iii) intensity-dependent sensor and environment noise $\mathbf{n} \in \mathbb{R}^{H \times W \times 3}$. We approximate the mapping $\tilde{\mathcal{A}} : \mathbf{p} \mapsto \mathbf{s}$ using the following model:

$$\mathbf{s}(u, v) = C[\mathbf{p}\big(H^{-1}(u, v)\big)] + \mathbf{n}(u, v), \tag{5}$$

where $(u, v)$ represents spatial coordinates of each pixel in the sensor image and the operator $C[\cdot]$ performs a matrix-vector operation on the three color channels.

**Algorithm 1** Optimization procedure of Transform-triggered adversarial examples (digital space)

**Require:** Clean image $x$, model $f$, transform $\tau$, parameter-target pairs $\{(\bar{\theta}_i, y_i^*)\}$;
1: Initialize $\delta \leftarrow 0$;
2: **while** not converged **do**
3:     **for** each pair $i$ **do**
4:         Sample $\theta_i \sim \mathcal{U}_r(\bar{\theta}_i)$
5:     **end for**
6:     $L \leftarrow \sum_i \mathcal{L}\big(f(\tau(x+\delta;\theta_i)), y_i^*\big)$   ▷ Eqn. (3)
7:     $\delta \leftarrow \delta - \alpha \cdot \text{sign}(\nabla_\delta L)$   ▷ PGD step
8:     $\delta \leftarrow \text{clip}(\delta, -\varepsilon, \varepsilon)$
9: **end while**
10: **return** $\delta$

**Algorithm 2** Optimization procedure of Transform-triggered adversarial examples (physical space)

**Require:** Clean image $x$, model $f$, transform $\tau$, pairs $\{(\bar{\theta}_i, y_i^*)\}$, approximated model $\tilde{\mathcal{A}}$;
1: Initialize $\delta \leftarrow 0$, $\mathbf{p} \leftarrow \texttt{Display}(\tau(x+\delta;\theta_i))$;
2: **while** not converged **do**
3:     **for** each pair $i$ **do**
4:         Sample $\theta_i \sim \mathcal{U}_r(\bar{\theta}_i)$
5:     **end for**
6:     $\mathbf{s} \leftarrow \texttt{Capture}(\mathbf{p})$   ▷ Sensor image
7:     $L \leftarrow \sum_i \mathcal{L}\big(f(\mathbf{s}), y_i^*\big)$   ▷ Aggregate losses
8:     $\delta \leftarrow \delta - \alpha \cdot \text{sign}(J_\tau^\top J_{\tilde{\mathcal{A}}}^\top(\mathbf{p})\nabla_{\mathbf{s}} L)$   ▷ PGD step
9:     $\delta \leftarrow \text{clip}(\delta, -\varepsilon, \varepsilon)$
10:     $\mathbf{p} \leftarrow \texttt{Display}(\tau(x+\delta;\theta_i))$   ▷ Planar scene
11: **end while**
12: **return** $\delta$

Standard physical attacks in (Athalye et al., 2018; Thys et al., 2019) perform complete optimization in the digital space, using transform as augmentations, and then evaluate under physical environment distortion. We perform optimization with the hardware-in-the-loop, which closes the gap between purely digital attacks and physically realizable adversarial examples. The model in equation 5 offers a differentiable approximation $\tilde{\mathcal{A}}$, where the gradient of the perturbation can be represented as $\nabla_\delta \mathcal{L} = J_\tau^\top J_{\tilde{\mathcal{A}}}^\top(\mathbf{p}) \nabla_{\mathbf{s}} \mathcal{L}$, $\nabla_{\mathbf{s}} \mathcal{L} = \left(\frac{\partial f}{\partial \mathbf{s}}\right)^\top \frac{\partial \mathcal{L}}{\partial f}$. Notably, the physical forward pass $\mathbf{s} = \mathcal{A}(\mathbf{p})$ introduces hardware-memory disconnection (i.e., the captured scene $\mathbf{s}$ retains no computational graph to the input $x$). We therefore compute the gradient in two parts that are stitched at the sensor/scene boundary: (i) $\nabla_{\mathbf{s}} \mathcal{L}$, obtained by `autograd` (a common operator for gradient computation in modern machine learning packages (Paszke et al., 2017)) through the network $f$ on the captured image $\mathbf{s}$; and (ii) the analytic surrogate Jacobian $J_\tau^\top J_{\tilde{\mathcal{A}}}^\top(\mathbf{p})$, evaluated in closed form on the displayed $\mathbf{p}$ from equation 5. This formulation closes the gap between digital attacks and physically realizable adversarial examples. The step-by-step optimization procedures for the physical space is in algorithm 2.

The hardware-in-the-loop formulation provides a more challenging test compared to purely digital attacks. The observation that transform-conditioned adversarial examples survive a real acquisition pipeline indicates that the perturbation encodes genuine transform-conditioned behavior, rather than merely showing digital artifacts. This observation has further implications for defenses that consider preprocessing operations such as resizing, compression, denoising, gamma correction, stabilization, and zoom as benign or as components of input-transform defenses. Our formulation and experiments show that such transforms can act as implicit triggers when the attacks are optimized conditionally over the transform space. Defenses should therefore test not only average robustness under transformed inputs, but also whether different preprocessing regimes induce systematic behavioral switching. Randomization helps only when it pushes inputs outside the attacker-specified transform neighborhoods or destroys the transform-conditioned partition. If the attacker optimizes over the same range, the randomization is absorbed into the trigger distribution.

## 4 Experiments

In this section, we first evaluate transform-triggered adversarial examples across diverse image transforms and model architectures. We then demonstrate the generalization of our formulation to object detection, highlighting its practical relevance. Finally, we extend our results from digital to a physical display–camera pipeline, showing that multi-targeted attack behaviors remain robustly partitioned under real-world noise and capture artifacts.

### 4.1 Experiment setup

**Models and dataset.** We utilize pretrained image classification models from Pytorch Torchvision (Paszke et al., 2017), which provides models pretrained on ImageNet dataset (Deng et al., 2009) for a vari-

Table 2: Transform-dependent targeted ASR (%) ↑ against classifiers over the range of selected parameters. A higher ASR value indicates better attack performance. The perturbation budget is $\varepsilon = 8$.

| Transform parameter | Classifier model | | | | | | | |
|---|---|---|---|---|---|---|---|---|
| | VGG19 | ResNet50 | Dense121 | Incv3 | Mobv2 | ViT-L16 | ViT-L32 | Swin-T |
| $S \sim [0.4, 0.6]$ | 95.80 | 87.20 | 88.20 | 61.90 | 96.40 | 68.70 | 61.00 | 99.70 |
| $S \sim [0.9, 1.1]$ | 99.90 | 98.70 | 98.20 | 83.10 | 100.0 | 75.50 | 68.40 | 100.0 |
| $S \sim [1.4, 1.6]$ | 99.90 | 99.60 | 99.60 | 80.30 | 99.80 | 75.90 | 68.70 | 100.0 |
| Average | 98.53 | 95.17 | 95.33 | 75.10 | 98.73 | 73.37 | 66.03 | 99.90 |
| $\sigma \sim [0.4, 0.6]$ | 100.0 | 99.90 | 99.80 | 95.40 | 99.80 | 97.40 | 91.70 | 100.0 |
| $\sigma \sim [1.4, 1.6]$ | 98.70 | 98.90 | 97.90 | 75.40 | 95.80 | 77.60 | 66.20 | 98.90 |
| $\sigma \sim [2.9, 3.1]$ | 97.10 | 98.90 | 94.40 | 65.70 | 94.20 | 65.60 | 53.80 | 99.10 |
| Average | 98.60 | 99.23 | 97.37 | 78.83 | 96.60 | 80.20 | 70.57 | 99.33 |
| $\gamma \sim [0.4, 0.6]$ | 99.90 | 100.0 | 99.90 | 93.10 | 99.90 | 99.00 | 93.30 | 99.80 |
| $\gamma \sim [0.9, 1.1]$ | 100.0 | 96.20 | 99.90 | 95.10 | 99.90 | 97.60 | 91.90 | 100.0 |
| $\gamma \sim [1.9, 2.1]$ | 100.0 | 91.00 | 99.70 | 93.70 | 99.70 | 93.60 | 85.30 | 99.70 |
| Average | 99.97 | 95.73 | 99.83 | 93.97 | 99.83 | 96.73 | 90.17 | 99.83 |
| $Q \sim [19, 21]$ | 96.10 | 87.70 | 91.70 | 73.40 | 86.90 | 66.60 | 72.70 | 83.40 |
| $Q \sim [49, 51]$ | 99.10 | 96.20 | 96.70 | 80.90 | 95.20 | 81.90 | 83.00 | 96.20 |
| $Q \sim [79, 81]$ | 99.40 | 98.40 | 99.20 | 87.00 | 97.90 | 87.80 | 82.20 | 99.80 |
| Average | 98.20 | 94.10 | 95.87 | 80.43 | 93.33 | 78.77 | 79.30 | 93.13 |

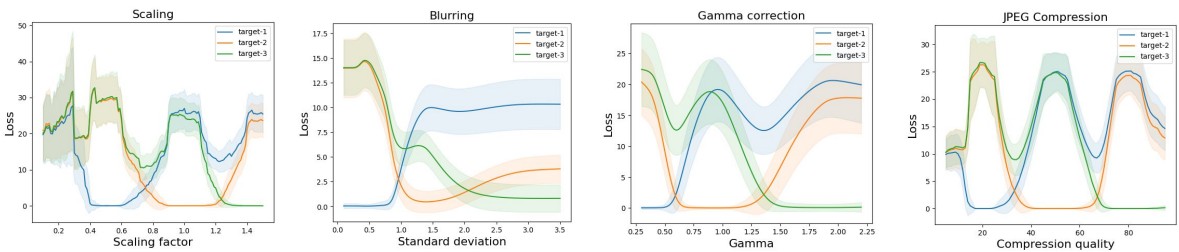

Figure 2: Loss landscape of the `ResNet50` over transform parameter values. Small loss values indicate successful targeted attacks within the corresponding transform parameter ranges. The shaded regions represent the standard deviation of each curve. Perturbations are generated to make the model predict three target labels for three transform parameter ranges: $S \sim \{[0.4, 0.6], [0.9, 1.1], [1.4, 1.6]\}$, $\sigma \sim \{[0.4, 0.6], [1.4, 1.6], [2.9, 3.1]\}$, $\gamma \sim \{[0.4, 0.6], [0.9, 1.1], [1.9, 2.1]\}$, $Q \sim \{[19, 21], [49, 51], [79, 81]\}$, same as Table 2. This figure indicates that 1) different transforms have various smoothness for the transform-triggered objective, 2) multiple attack targets can be controllably triggered by desired transform parameters, and 3) attacks remain effective when the parameter is sampled outside of the ranges used in optimizations.

ety of families and architectures. Models are trained with data augmentation techniques including random cropping, rotation, flipping, and color jittering, so that they align with our assumption in equation 1. We sample models from different families that cover Convolutional Neural Networks (CNNs) and Vision Transformers (ViT): {`VGG-19-BN`, `ResNet-50`, `DenseNet-121`, `InceptionV3`, `MobileNet-v2`, `ViT-L-16`, `ViT-L-32`, `Swin-T`}. We use 1000 ImageNet-like RGB images from the NeurIPS'17 challenge (Brain, 2017), which has the same label space as ImageNet, all images have size $224 \times 224$.

**Our threat model and attack setup.** We consider targeted, transform-triggered adversarial attacks, which are more challenging than untargeted ones and directly align with our formulation. The attacker is assumed to have whitebox access to the victim model during optimization and knowledge of sets of transforms that input may undergo in real model deployment. The goal is to induce distinct, attacker-specified target labels conditioned on these transforms when the perturbed image is processed under corresponding transform parameters. In the main paper, we evaluate a simplified setup optimizing $N = 3$ transform-target pairs $\{\bar{\theta}_i\}_{i=1}^3$ and $\{y_i^\star\}_{i=1}^3$, where target labels are randomly sampled from ImageNet. We provide stress tests for

larger $N$ in section 5.1. Optimization is performed using PGD with 500 iterations and step size $\alpha = 5 \times 10^{-4}$, which we found sufficient for convergence.

**Evaluation metrics.** We evaluate the effectiveness of transform-triggered adversarial examples using the Attack Success Rate (ASR). Specifically, for a perturbed image transformed as $\tau(\mathbf{x} + \delta; \theta_i)$, we check whether the prediction matches target label $y_i^*$ (i.e., success if $f(\tau(\mathbf{x} + \delta; \theta_i)) = y_i^*$). We report ASR for each transform parameter $\bar{\theta}_i$ over $\mathcal{U}_r(\bar{\theta}_i)$, where $\mathcal{U}_r(\bar{\theta}_i)$ includes series of $\theta$ samples generated with a small sampling rate (detailed in section 4.2), and we report ASR averages over all samples within $\mathcal{U}_r(\bar{\theta}_i)$ for each $\bar{\theta}_i$.

## 4.2 Attacks against classifiers

We present transform-triggered examples under four image transforms commonly applied in real-world scenarios: geometric transforms (scaling, blurring) and photometric transforms (gamma correction, JPEG compression). For each transform, we select three parameters $\{\bar{\theta}_i\} \in \Theta_{\text{trigger}}$ following **two concrete criteria:** (i) covering a representative spectrum of deployment conditions: unaltered, mildly transformed, and heavily transformed inputs (illustrated qualitatively in Fig. 1); and (ii) preserving model accuracy on clean samples at each $\bar{\theta}_i$ so that the equation 1 assumption holds and observed adversarial behavior is attributable to the objective equation 3 rather than transform-induced degradation to the model ( accuracies on transformed clean examples are reported in Tab. 12, Appendix C).

For scaling, we use factors $S \in \{0.5, 1.0, 1.5\}$. For blurring, we fix the Gaussian kernel size to $5 \times 5$ and vary $\sigma \in \{0.5, 1.5, 3.0\}$. For gamma correction, we select $\gamma \in \{0.5, 1.0, 2.0\}$, and for JPEG compression, we use quality levels $Q \in \{20, 50, 80\}$. To generate $\mathcal{U}_r(\bar{\theta}_i)$, we define a neighborhood with an interval radius of $r = 0.1$ for scaling, blurring, gamma correction, and $r = 1$ for JPEG compression. This results in the following parameter ranges: scaling $S \sim \{[0.4, 0.6], [0.9, 1.1], [1.4, 1.6]\}$, blurring $\sigma \sim \{[0.4, 0.6], [1.4, 1.6], [2.9, 3.1]\}$, gamma correction $\gamma \sim \{[0.4, 0.6], [0.9, 1.1], [1.9, 2.1]\}$, and JPEG compression $Q \sim \{[19, 21], [49, 51], [79, 81]\}$.

Tab. 2 summarizes the ASR for transform-triggered attacks across three target labels and corresponding parameter ranges. These results show consistent targeted attack success on whitebox CNN and ViT models, demonstrating the effectiveness of transform-triggered adversarial examples. These attacks, optimized via equation 3, retain their effectiveness under small parameter variations (also indicated in Fig. 2). Their transferability to blackbox and defended models is further explored in section A.1. Fig. 2 visualizes the adversarial loss landscape of `ResNet50` across four transforms, with loss evaluated over a grid of transform parameters (sampling rate 0.1 for $S, \sigma, \gamma$, and 1 for $Q$). The three colors represent distinct target labels, with solid lines indicating average loss and shaded areas denoting standard deviation. As seen, minimum loss values align with the intended transform parameter ranges, confirming that adversarial examples successfully embed targeted attacks triggered by specific transformations. Notably, scaling and JPEG are more sensitive to parameter variations, while blurring and gamma correction exhibit greater smoothness.

## 4.3 Attacks against object detectors

To demonstrate the practicality of transform-triggered examples, we design a transform-selective hiding threat model compatible with our objectives. We present more threat models against detectors in Sec. E. Here, we first outline the setups, followed by a discussion of the threat model and results.

**Models and dataset.** For object detection, we use pretrained models from `MMDetection` (Chen et al., 2019) on the COCO 2017 dataset (Lin et al., 2014), and generate adversarial examples using the same dataset. Our model selection covers diverse architectures: one-stage detector (`YOLOv3` (Redmon & Farhadi, 2018)), two-stage detectors (`Faster R-CNN` (Ren et al., 2015), Grid R-CNN (Lu et al., 2019)), and a ViT-based model (DETR (Carion et al., 2020)).

**Threat model: transform-selective hiding.** In this threat model, we maintain the same attacker knowledge assumptions described in section 4.1, aim to hide all detectable objects when a level of image enhancement is applied that potentially discloses sensitive information, and preserve detectability in unaltered or minimally enhanced images (consistent criteria as in section 4.2), enabling privacy protection and strategic information control. For instance, surveillance footage can obscure sensitive objects upon enhancement, preventing unauthorized inspection while maintaining visibility in unaltered conditions. Similarly,

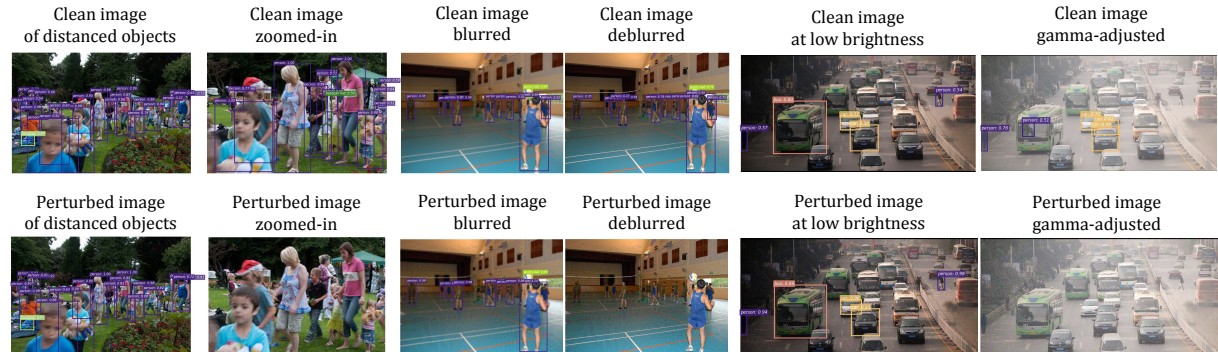

Figure 3: Visualization of the enhancement transform-triggered hiding attacks on YOLOv3. While objects in the enhanced clean images are being detected, after adding transform-triggered perturbations (with budget $\varepsilon = 10$), detectors fail when the enhancement transform is applied, motivating its utility for preventing sensitive information disclosure in remote sensing or public surveillance systems.

Table 3: Transform-selective hiding attack success rate (ASR) for trigger zone and accuracy (ACC) for safe zone. Higher ASR and ACC indicate better performance. The attack is selectively triggered within the desired range of enhancement transform parameters. Our attack consistently hides objects under image enhancement transforms while preserving detectability in unaltered or minimally enhanced images.

| $\Theta$ | Zoom-in | | | | Deblurring | | | | Gamma correction | | | |
|---|---|---|---|---|---|---|---|---|---|---|---|---|
| | Faster | YOLOv3 | GRID | DETR | Faster | YOLOv3 | GRID | DETR | Faster | YOLOv3 | GRID | DETR |
| $\Theta_{\text{trigger}}$ | 89.41 | 95.22 | 84.62 | 53.79 | 78.92 | 85.17 | 72.37 | 51.03 | 79.88 | 89.10 | 72.13 | 53.46 |
| $\Theta_{\text{safe}}$ | 94.36 | 99.03 | 93.25 | 77.02 | 92.43 | 91.78 | 88.84 | 71.53 | 89.49 | 91.78 | 82.50 | 65.96 |

in satellite imaging, critical targets can be concealed under specific enhancements, controlling detectability based on operational needs. We stress that transform-selective hiding is a representative instantiation: our formulation more broadly supports transform-conditioned behaviors depending on the security needs.

**Results.** We consider attacking zoom-in (i.e., scaling + centered cropping) with $S \sim [2.0, 2.5]$, blurring with $\sigma \sim [0.0, 0.7]$, and gamma correction with $\gamma \sim [0.5, 0.9]$ to simulate these enhancement scenarios. Outside of attack ranges, which we consider as safe ranges, we keep the detectability of objects and report accuracy. Tab. 3 shows that the attack is selectively triggered within the desired range of enhancement transform parameters, with overall ASR above 50% over attack ranges and high ACC over safe ranges. Our attack consistently hides objects when different image enhancement transforms are heavily applied, while preserving detectability in unaltered or minimally enhanced images. The examples in Fig. 3 show that objects are selectively hidden in predefined attack ranges in three enhancement transforms, meanwhile, over safe range the detection results remain identical to original clean image. We provide animated visual examples of these attack scenarios in the submitted supplementary material.

## 4.4 Hardware-in-the-loop experiments

In this section, we further evaluate transform-triggered adversarial examples in a physical setup created using a planar display and a DSLR camera. Our goal is to verify that a single perturbation can reliably induce multiple adversarial behaviors under different transforms in physical attacks and that the transform-triggered behaviors are desired and targeted rather than precision artifacts introduced by transform functions. As described in equation 5, we represent the imaging model using a combination of homography, color correction, and noise calibration. We include the diagram of our hardware-in-the-loop attack pipeline, and a picture of the built physical setup, in Fig. 4.

**Implementation details.** We display perturbed images on a `BenQ BL2780 27" 1080p (1920×1080)` monitor and capture them using a Nikon Z30 camera (lens: `NIKKOR Z DX 16-50mm f/3.5-6.3 VR`; manual mode, aperture `f/5.6`, ISO 100, shutter 1/60 s; working distance: ∼60 cm). The captured images are then fed to the model loaded in memory.

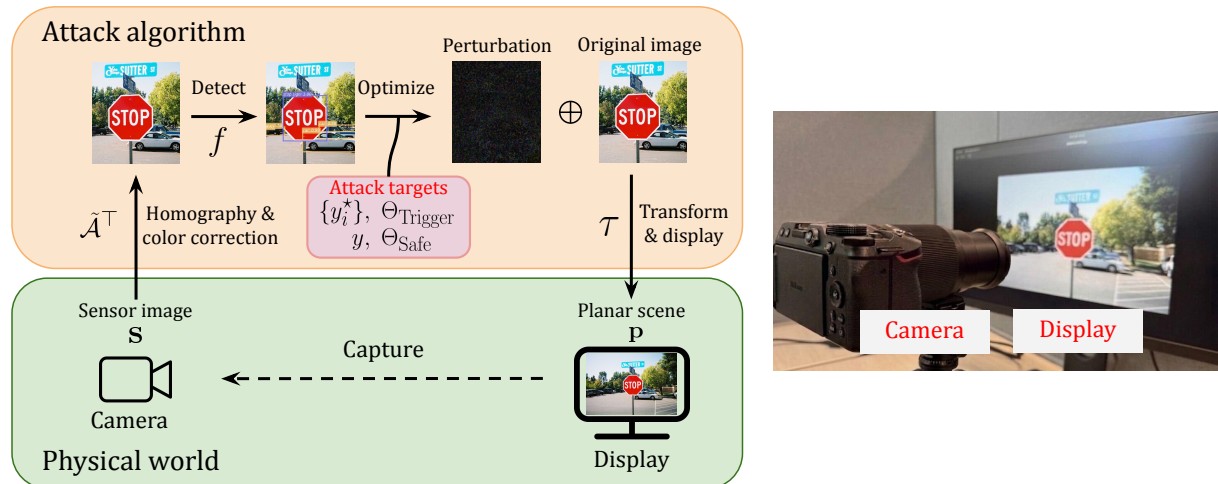

Figure 4: The diagram of hardware-in-the-loop pipeline and a picture of our built camera-display setup. In the physical world, we display transformed adversarial example on a planar display, then capture the adversarial example with a DSLR camera. The captured images are then fed to the model loaded in the memory, optimized using the transform-triggered adversarial-example formulation.

We estimate the homography matrix, $H$, by registering four markers at image corners across physical and captured digital images. We estimate $C$ using 1000 pairs of RGB vectors, randomly sampled from the matching pixels in the images captured by the camera and the corresponding values stored in the memory.

**Image classification.** We follow the same experimental setup as in Sec. 4.2, sampling 100 images from ImageNet-1K and four pretrained classifiers from `TorchVision`. For each image, we assign one target label to the original view and a different target label to the transformed version, optimizing two targeted attacks per instance for Zoom (zoom factor $\{1.0, 1.2\}\times$), Blur (Gaussian kernel $\sigma \in \{0.1, 1.0\}$), and Gamma ($\gamma \in \{0.7, 1.0\}$). In Tab. 4, we first report clean classification accuracy on non-perturbed images and the ASR of standard PGD (a single fixed target, no transform), which serve as references for our physical pipeline under benign and conventional threats, respectively. We then report the ASR of transform-triggered examples, computed as the average targeted success across the two transforms optimized per instance. The results show that (i) standard PGD attains near-100% ASR despite clean accuracy of only 69–83%, confirming that the physical pipeline transmits targeted perturbations reliably even where benign inputs are already misclassified; and (ii) the transform-partitioned attacks remain robust under physical distortion, incurring only 4.15–5.5% average ASR degradation relative to single-target PGD.

Table 4: Physical attacks on image classifiers using transform-triggered adversarial examples across three image transforms. Our proposed transform-triggered formulation creates robust and physically realizable adversarial examples.

| Model | Classification Accuracy (↑) | Attack Success Rate (↑) | | | |
|---|---|---|---|---|---|
| | | PGD | Zooming | Blurring | Gamma |
| VGG19 | 69.00 | 100.0 | 96.00 | 95.00 | 96.00 |
| ResNet50 | 73.00 | 99.00 | 95.00 | 96.00 | 92.00 |
| Dense121 | 69.00 | 100.0 | 98.00 | 99.00 | 98.00 |
| ViT-L16 | 83.00 | 100.0 | 94.00 | 93.00 | 91.00 |

**Object detection.** We further evaluate a real-world zoom-triggered attack on `Faster R-CNN` using stop sign images from COCO. We learn additive perturbation, $\delta$, that is localized within the stop sign region and optimized to produce two mutually incompatible outcomes: (i) original to minor zoom ($1.0$–$1.4\times$ as $\Theta_{\text{safe}}$) should detect the stop sign; (ii) moderate to strong zoom ($1.5$–$2.0\times$ as $\Theta_{\text{trigger}}$) should trigger the hiding attack (i.e., not detect the stop sign). To emulate real physical zoom rather than digital cropping, we display the perturbed image and change the distance between the camera and the monitor. Representative captured

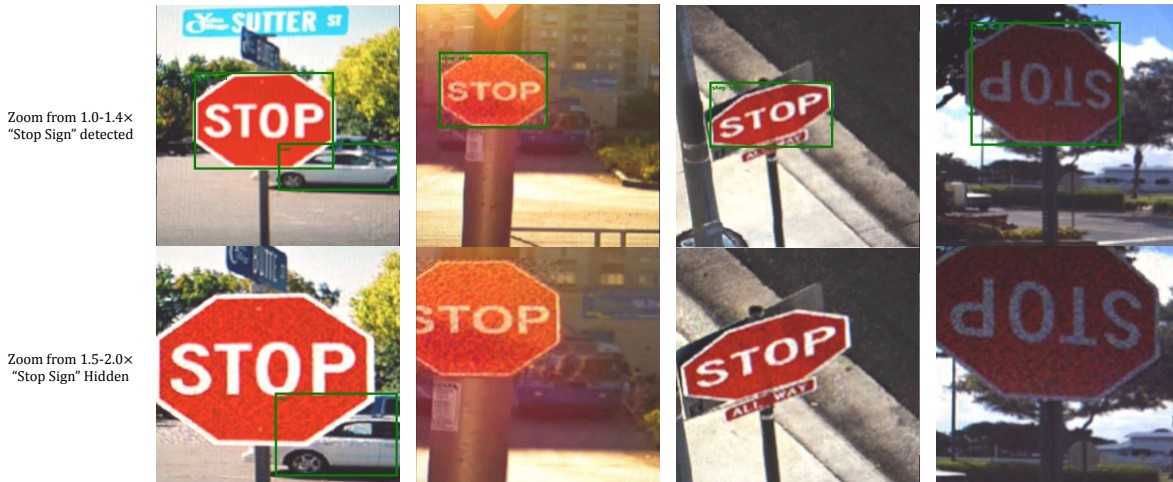

Figure 5: Physical attacks on the `Faster R-CNN` object detector using a zoom trigger. Adversarial examples are displayed on a monitor and captured by a DSLR camera. Top row: captures under "safe zoom," where objects are correctly detected. Bottom row: captures under "trigger zoom," where the targeted attack activates and the target object (stop sign) is hidden.

frames are shown in Fig. 5 and videos are provided in the supplementary material. The detector outputs follow the transform-triggered logic enforced by optimization: the stop sign is detected in the $1 - 1.4\times$ zoom range; whereas, all proposals are suppressed in the $1.5 - 2.0\times$ zoom range and the stop sign is not detected by the model. Note that "car" is detected at all zoom levels, which is the desired attack effect in our setup. These results show that the same physical object can be made to yield different model outputs depending solely on the applied transform, demonstrating physically realistic transform-triggered attacks on object detection, which is a substantially harder setting than classification.

## 5 Discussion

### 5.1 Capacity limit of a single perturbation

Given that our attack can embed multiple transform-dependent targeted behaviors into a single perturbation, a natural question is: *"How many transform-triggered targets can one perturbation support?"* To empirically probe this limit, we conduct a stress test with up to $N = 25$ targets on three representative models from section 4.2: `ResNet50, InceptionV3` and `ViT-L-16`. We use scaling, blurring, gamma correction, and JPEG compression as transform triggers. For each transform, we initialize the parameter sequence with $\theta_1 = 0.5$ for scaling, blurring, and gamma correction, and $\theta_1 = 20$ for JPEG compression. Additional trigger parameters are then appended with an adaptive step size until reaching $N = 25$. Each newly added parameter $\theta_i$ is assigned a randomly sampled ImageNet target label $y_i^\star$. This stress test imposes $N$ transform-conditioned targets constraint on one bounded perturbation:

$$f(\tau(x + \delta; \theta_i)) = y_i^\star, \qquad i = 1, \ldots, N, \qquad \|\delta\|_p \leq \varepsilon. \tag{6}$$

Given an evaluation set $\mathcal{D}$, we summarize performance by the empirical average targeted ASR:

$$\text{ASR}_N(\delta) = \frac{1}{N|\mathcal{D}|} \sum_{i=1}^{N} \sum_{x \in \mathcal{D}} \mathbf{1}[f(\tau(x + \delta; \theta_i)) = y_i^\star]. \tag{7}$$

Thus, increasing $N$ imposes more target constraints on the same perturbation under fixed $\varepsilon$. As Fig. 6 shows, the average targeted ASR decreases with $N$, but the degradation rate depends strongly on the transform and victim architecture: scaling and JPEG compression sustain higher ASR under larger $N$, while blurring and gamma correction degrade faster. This is consistent with the loss landscapes in Fig. 2:

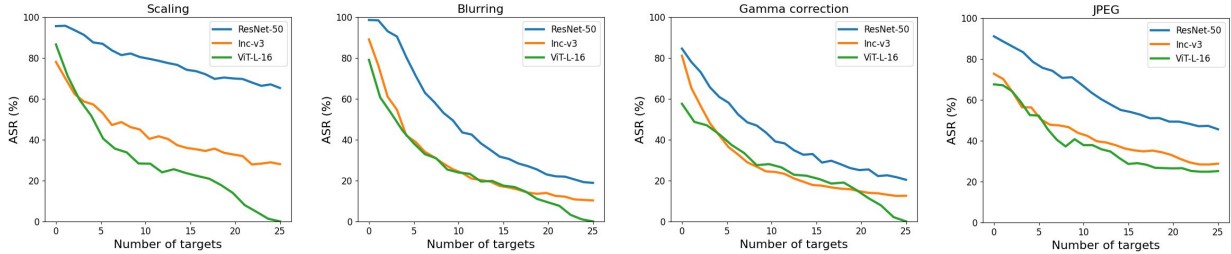

Figure 6: Average targeted ASR vs the number of embedded targets $N$ across scaling, blurring, gamma correction, and JPEG compression. As $N$ increases, the average ASR drops for fixed perturbation budget.

sharper, more separable losses allow finer partitioning of the transform space, whereas smoother losses create wider overlapping trigger regions and fewer distinguishable targets. Architecture is also a factor: `ResNet50` accommodates more transform-target pairs, while `ViT-L-16` shows smaller capacity, indicating that capacity also depends on the model-dependent decision-boundary geometry under transformed inputs. Finally, the budget $\varepsilon$ acts as a direct constraint: larger $\varepsilon$ supports more targets at the cost of increased visibility.

Overall, the empirical capacity of a single perturbation depends jointly on: number of targets $N$, smoothness of transform functions, victim architecture, and perturbation budget. A formal capacity theory relating these factors to the maximum number of embeddable targets remains an important direction for future work.

## 5.2 Practical threat scenarios

Transform-triggered adversarial examples suggest a practical threat model for vision systems with known or partially known processing pipelines. In real deployments, model inputs are typically produced by an image signal processing (ISP) pipeline involving transforms such as denoising, gamma correction, resizing, and compression, which are often deterministic or configuration-dependent once the camera model, software stack, or deployment setting is fixed. An attacker with knowledge of this processing pipeline could thus optimize a perturbation whose malicious behavior is conditionally activated under targeted ISP transforms while remaining benign under others. Prior work partially supports this threat model in the visible-patch regime. TPatch (Zhu et al., 2023) studies an active trigger in which an attacker injects an acoustic signal into a vehicle MEMS gyroscope, causing the Optical Image Stabilization (OIS) module to apply enhancement transforms that activate the adversarial patch. Our work generalizes this: ordinary image transforms ubiquitous in sensing and preprocessing pipelines can themselves act as implicit triggers and be exploited to inject multiple malicious behaviors. This makes transform-triggered adversarial examples relevant for evaluating robustness across real-imaging scenarios: heterogeneous cameras, compression settings, and preprocessing configurations, which are dynamic conditions that conventional adversarial examples cannot capture.

## 6 Conclusion

We introduce a framework for transform-triggered adversarial examples that expose a previously uncharacterized dynamical vulnerability in deep networks. Conventional adversarial examples induce a single, static failure mode. Our method can generate adversarial examples that can produce multiple, targeted adversarial behaviors triggered by image transforms. Extensive experiments across architectures and tasks show that these attacks enable precise, transform-dependent mispredictions. Our hardware-in-the-loop experiments further confirm that the adversarial examples remain effective in physical attacks, indicating a robust feature-space phenomenon beyond digital artifacts. Our proposed method can potentially be beneficial for sensitive operations where perturbations remain benign under normal viewing and imaging conditions but suppress sensitive content under adverse conditions, enabling transform-triggered privacy or safety control.

## Acknowledgment

This work was supported in part by an NSF award CCF-2046293, a UCR OASIS award, and a UC SoCal HUB seed award.

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

## Appendix

In this appendix, we provide several additional analyses and supporting results. In section A, we present additional evaluations of transform-triggered adversarial examples, including blackbox transferability, adaptation to alternative optimization methods, and more challenging target-label selections. In section B, we extend transform-triggered adversarial examples to additional image transforms, namely perspective/viewpoint changes and flipping, beyond those discussed in the main paper. In section C, we report the accuracy of the classifier models used in our main experiments (section 4.2) under input transforms, providing a reference for model accuracy and supporting the assumption in equation 1. Finally, in section D, we report the computational resources used in our experiments, and in sections E and F, we provide additional visual examples of transform-triggered adversarial examples on classifiers and detectors.

## A    Additional evaluation

In section A.1, we first provide a blackbox transfer test of adversarial examples generated by our transform-conditioned, multi-target formulation. In section A.2, we then conduct an ablation for the adversarial objectives, showing that the image-transform factor in the objective, not the multi-target component, drives blackbox transferability. In section A.3, we benchmark alternative optimization algorithms beyond PGD for generating transform-triggered adversarial examples, supporting that PGD is the most effective choice. Additionally, we evaluate attacks on more challenging label selection in section A.4.

### A.1 Transfer test on blackbox and defended models

**Blackbox transferability.** In our formulation, post-optimized transform-triggered adversarial examples become a function of transform parameters (i.e., $x^\star = \tau(x + \delta; \theta)$), enabling query-efficient blackbox attacks. Since only model outputs are accessible in the blackbox setup, we query blackbox models with transformed versions of perturbed images $\tau(x + \delta; \theta)$ over $\mathcal{U}_r(\bar{\theta}_i)$, with three queries per image at maximum, which is a negligible query cost. For an adversarial example $x + \delta$ generated for a surrogate model, we sample transform parameters $\theta$ from the neighborhoods used in Tab. 2, i.e., $\mathcal{U}_r(\bar{\theta}_i)$. The goal is to find a transform parameter $\theta_i^\star$ such that the adversarial example $\tau(x + \delta; \theta_i^\star)$ deceives the blackbox model:

$$\text{Find } \theta_i^\star \quad \text{s.t.} \quad f_{\text{bb}}(\tau(x + \delta; \theta_i^\star)) = y_i^\star; \ \theta_i^\star \in \mathcal{U}_r(\bar{\theta}_i). \tag{8}$$

At test time, we conduct a uniform search over $\mathcal{U}_r(\bar{\theta}_i)$ with a sampling interval of 0.1, leading to three queries per transform parameter (e.g., $\{0.4, 0.5, 0.6\}$ for $S = 0.5$). An attack is successful if a transformed adversarial example forces the blackbox model to predict the target label.

For untargeted attacks, we use the ground-truth label ($y$) as the target in equation 3 for all transform parameters $\theta_i$ and switch the objective from minimization to maximization. Because all $N$ targets collapse to the single $y$, the multi-target sum in equation 3 reduces to a single expectation over the union of transform neighborhoods, recovering the EOT objective (Athalye et al., 2018) as a special case of our formulation. An untargeted attack is successful when $\delta$-perturbed input to a blackbox model $f_{\text{bb}}$ satisfies:

$$\text{Find } \theta_i^\star \quad \text{s.t.} \quad f_{\text{bb}}(\tau(x + \delta; \theta_i^\star)) \neq y; \ \theta_i^\star \in \mathcal{U}_r(\bar{\theta}_i). \tag{9}$$

We present the transfer attack success rates (ASRs) under both targeted and untargeted setups, in Tab. 5. As a reference, we compile transfer attacks ASRs reported in BPA (Wang et al., 2023a), SU (Wei et al., 2023), and ILPD (Li et al., 2024), out of consideration that they share same choice of dataset, surrogate and victim models as our work. Notably, due to the formulation difference that our adversarial examples try to achieve multi-target attacks, whereas other existing work focusing on single-target, the targeted attacks evaluation protocol is inherently misaligned, and a fair comparison is non-trivial. The purpose of this study is not to showcase achieving superior performance in blackbox transfer research domain, but as a reference for transform-triggered adversarial examples transferability.

Our method achieves comparable ASR to the selected blackbox attacks without specialized adaptation. Scale-and blur-triggered attacks achieve high success in both targeted and untargeted setups. Scale-triggered adversarial examples exhibit slightly better transferability than blur-triggered ones. Targeted success is higher for larger scaling factors, while untargeted success occurs more frequently at smaller scaling factors. We provide an ablation study in section A.2 that disentangles the individual contributions of transform function and multi-target optimization to blackbox transferability.

**Attacks against defended models.** To further evaluate the effectiveness of the transform-triggered adversarial examples, we assess their performance on four defense methods: HGD (Liao et al., 2018), Randomized Smoothing (RS) (Cohen et al., 2019), JPEG compression (JPEG) (Guo et al., 2018) and NPR (Naseer et al., 2020). We follow the untargeted setup in a recent transfer attacks BPA (Wang et al., 2023a), generate whitebox attacks with perturbation budget $\varepsilon = 8$, against four defenses applied to the `ResNet50` model. In Tab. 6, we report these untargeted ASRs averaged over all the transform parameters. Our attack achieves overall better performance than BPA (Wang et al., 2023a) on the same benchmark. These results suggest that transform-triggered examples can bypass existing defense methods by leveraging the transform space.

Table 5: Blackbox transfer evaluation under untargeted and targeted settings, adversarial perturbation budget is $\varepsilon = 8$. Our scale-triggered attacks achieve comparable untargeted ASRs and higher targeted ASRs under the same setting as the most recent transfer attacks. Furthermore, the transfer attacks maintain transform-triggered attack properties as shown by the targeted ASRs.

| Methods | Untargeted ASR (%) ↑ | | | | | Targeted ASR (%) ↑ | | | | |
| | Surrogate | Blackbox model | | | | Surrogate | Blackbox model | | | |
| | ResNet50 | VGG19 | Dense121 | Incv3 | Mobv2 | ResNet50 | VGG19 | Dense121 | Incv3 | Mobv2 |
|---|---|---|---|---|---|---|---|---|---|---|
| BPA (Wang et al., 2023a) | 99.40 | 60.96 | 70.70 | 35.36 | 68.90 | 100.0 | 31.02 | 43.82 | 15.34 | 39.00 |
| ILPD (Li et al., 2024) | 83.96 | **88.10** | **90.68** | 64.70 | - | - | - | - | - | - |
| Logit-SU (Wei et al., 2023) | - | - | - | - | - | - | 41.30 | 45.70 | 1.10 | - |
| $S \sim [0.4, 0.6]$ | 88.00 | 80.10 | 81.10 | **96.90** | **86.80** | 99.60 | 39.10 | 40.00 | 12.10 | 31.50 |
| $S \sim [0.9, 1.1]$ | 98.30 | 86.70 | 77.50 | 62.60 | 81.00 | 99.90 | **62.80** | 61.40 | 28.80 | 56.10 |
| $S \sim [1.4, 1.6]$ | 99.90 | 85.40 | 83.90 | 58.90 | 85.10 | 100.0 | 59.50 | **67.80** | **33.10** | **60.00** |
| $\sigma \sim [0.4, 0.6]$ | 99.80 | 64.40 | 69.40 | 53.50 | 65.60 | 100.0 | 32.60 | 42.20 | 19.10 | 35.00 |
| $\sigma \sim [1.4, 1.6]$ | 99.80 | 85.40 | 84.60 | 73.70 | 83.80 | 99.90 | 43.90 | 52.60 | 23.40 | 35.40 |
| $\sigma \sim [2.9, 3.1]$ | 99.60 | 86.80 | 85.50 | 76.30 | 85.00 | 99.90 | 41.90 | 50.40 | 22.50 | 33.80 |

Table 6: Results of untargeted ASR (%) ↑ against defense methods, perturbation budget is $\varepsilon = 8$.

| Attack method | Defense method | | | |
| | HGD | RS | JPEG | NPR |
|---|---|---|---|---|
| BPA | 23.96 | 14.00 | 22.52 | 14.08 |
| Scaling (ours) | 56.20 | 53.43 | 34.90 | 39.70 |
| Blurring(ours) | 57.73 | 63.67 | 57.80 | 52.43 |
| Gamma(ours) | 48.67 | 65.43 | 52.27 | 53.57 |

## A.2 Ablation on multi-target objective transferability

In this section, we provide an ablation study answering the question: *does the transform-triggered objective with multiple targets yield more blackbox-transferable adversarial examples?* To disentangle the roles of the *transform function* and the *multi-target* objective, we formulate three distinct objectives and optimize a separate set of adversarial examples for each. Objective 1 (Obj. 1) corresponds to Eq. (3), where we assign $N = 3$ random targets $y_i^\star$ to parameters $\bar{\theta}_i$, yielding adversarial examples under a *transform with multi-target* formulation. Obj. 2 corresponds to EOT, solving $\max_\delta \sum_{i=1}^N \mathcal{L}\big(f(\tau(x + \delta; \theta_i)), y\big)$, as a *transform with single-target* formulation. Obj. 3 corresponds to $\min_\delta \sum_{i=1}^N \mathcal{L}(f(x + \delta), y_i)$, as a *multi-target without transforms* formulation. All objectives are subject to (s.t.) the perturbation budget constraint $\|\delta\|_\infty \leq 8$.

Following the untargeted setup in section A.1 that utilizes `ResNet50` as surrogate and evaluate the blackbox untargeted ASR (described in equation 9) with adversarial examples generated from solving Obj. 1,2 and 3. The ablation in Tab. 7 reveals that **transform function is the primary driver of transferability:** Obj. 3 (multi-target without transforms) consistently yields the lowest blackbox ASR, while Obj. 2 (transforms with a single target) performs comparably to or above Obj. 1. **This conclusion is consistent with findings in prior work that input diversity enhances blackbox transferability** (e.g., DIM (Xie et al., 2019), SIM (Lin et al., 2020), Admix (Wang et al., 2021)), and confirms that the transferability gains in Tab. 5 stem from transform-based input augmentation rather than the newly introduced multi-target formulation.

Importantly, this result is expected and does not conflict with the contribution of this work, as transferability and transform-triggered multi-target behavior are different properties of adversarial examples. Obj. 2 achieves marginally better transfer because it concentrates the entire perturbation budget on a single target, whereas Obj. 1 must partition that budget across $N$ targets and splitting the optimization effort across distinct targeted attacks. We again remind that contribution of this work is not in maximizing blackbox transfer but in identifying and characterizing a new whitebox phenomenon: embedding multiple conditionally-activated behaviors within one perturbation, which Obj. 2 cannot achieve by design. That Obj. 1 retains competitive transfer despite splitting its budget across targets further supports the transform-conditioned effects.

Table 7: Ablation study separating the effects of multi-target optimization and transform function on black-box transferability. We report untargeted ASR (%) for scale and blur transforms, with examples generated on the surrogate and evaluated on blackbox models. The transform-involved, single-target objective (Obj. 2, which is equivalent to EOT) yields the most transferable adversarial examples.

| Adversarial objective | | | | Untargeted ASR (%) ↑ | | | | |
| --- | --- | --- | --- | --- | --- | --- | --- | --- |
| | | | | Surrogate | Blackbox model | | | |
| Trigger | Obj. | Transform | Multi-target | ResNet50 | VGG19 | Dense121 | Incv3 | Mobv2 |
| Scale | 1 | ✓ | ✓ | 96.50 | 69.90 | 69.90 | 88.00 | 81.70 |
| | 2 | ✓ | ✗ | 95.40 | **84.00** | **80.83** | **88.10** | **84.30** |
| | 3 | ✗ | ✓ | 96.70 | 63.40 | 64.70 | 86.80 | 75.60 |
| Blur | 1 | ✓ | ✓ | 99.50 | 64.50 | 72.90 | 72.80 | 74.20 |
| | 2 | ✓ | ✗ | 99.70 | **78.87** | **79.83** | **67.80** | **78.10** |
| | 3 | ✗ | ✓ | 99.70 | 64.20 | 66.80 | 69.60 | 70.20 |

## A.3 Adaptation with other optimization algorithms

In main paper, we primarily solve transform-triggered adversarial examples using PGD (Madry et al., 2018) considering its simplicity and effectiveness. Yet in principle, these attacks can be generated with other optimization methods. Here, we evaluate the adaptability of transform-triggered adversarial examples using commonly used methods, including FGSM (Goodfellow et al., 2015), MIM (Dong et al., 2018), and C&W (Carlini & Wagner, 2017), and compare their performance with PGD.

Tab. 8 presents the average ASR for scale-dependent targeted attacks under the same settings as Tab. 2 in the main text. Among tested optimization algorithms, MIM, C&W, and PGD are sufficient to solve equation 3 and achieve high ASR, while the single-step FGSM is insufficient.

Table 8: Average scale-triggered ASRs (%) ↑ with different attack optimization methods, perturbation budget $\varepsilon = 8$. PGD offers overall better ASR among all optimization methods.

| Attack method | ResNet50 | VGG19 | Dense121 | Incv3 |
| --- | --- | --- | --- | --- |
| FGSM (Goodfellow et al., 2015) | 0.13 | 0.13 | 0.07 | 0.10 |
| C&W (Carlini & Wagner, 2017) | 85.90 | 90.73 | 90.77 | 60.50 |
| MIM (Dong et al., 2018) | 92.80 | 96.77 | 94.83 | **83.10** |
| PGD (Tab. 2) | **96.06** | **98.83** | **97.10** | 82.08 |

## A.4 More challenging attack targets

In section 4.2 of the main text, we initially employed a random selection process to choose three distinct classes from the set of 1000 ImageNet classes as our target labels, denoted as $\{y_i^\star\}_{i=1}^3$. Here, we maintain the consistent setup as section 4 (i.e., datasets, models, hyperparameters), and select the ***three least-likely labels*** extracted from the probability vectors converted from logits.

In this continuation, we evaluate the Adversarial Success Rate (ASR) under these modified attack settings, as presented in Tab. 9. This evaluation demonstrates the efficacy of our attack formulation even when faced with the challenge of targeting the least-likely labels.

Table 9: ASR evaluation of transform-triggered attacks using the three least-likely labels as targets (challenging selection). Higher value indicates better attack performance. The perturbation budget is $\varepsilon = 8$.

| Transform parameter | Classifier model ASR(%) ↑ | | | | | | | |
|---|---|---|---|---|---|---|---|---|
| | VGG19 | ResNet50 | Dense121 | Incv3 | Mobv2 | ViT-L-16 | ViT-L-32 | Swin-T |
| $S = 0.5$ | 98.60 | 89.60 | 92.10 | 77.70 | 96.20 | 81.70 | 65.40 | 99.40 |
| $S = 1.0$ | 100.0 | 99.70 | 99.70 | 91.00 | 100.0 | 95.40 | 85.80 | 100.0 |
| $S = 1.5$ | 100.0 | 99.40 | 99.00 | 82.50 | 99.80 | 91.80 | 78.00 | 100.0 |
| Average | 99.53 | 96.23 | 96.93 | 83.73 | 98.67 | 89.63 | 76.40 | 99.80 |
| $\sigma = 0.5$ | 100.0 | 99.80 | 99.90 | 90.60 | 99.80 | 94.40 | 88.70 | 100.0 |
| $\sigma = 1.5$ | 99.30 | 96.90 | 98.20 | 77.70 | 95.80 | 74.80 | 65.60 | 98.90 |
| $\sigma = 3.0$ | 99.40 | 95.60 | 98.20 | 74.50 | 94.20 | 71.20 | 55.90 | 99.10 |
| Average | 99.57 | 97.43 | 98.77 | 80.93 | 96.60 | 80.13 | 70.07 | 99.33 |
| $\gamma = 0.5$ | 100.0 | 99.60 | 99.80 | 91.90 | 99.90 | 98.40 | 90.90 | 99.80 |
| $\gamma = 1.0$ | 100.0 | 99.90 | 100.0 | 90.70 | 99.90 | 94.40 | 88.00 | 99.70 |
| $\gamma = 2.0$ | 100.0 | 99.60 | 99.70 | 89.10 | 99.70 | 85.60 | 76.80 | 99.90 |
| Average | 100.0 | 99.70 | 99.83 | 90.57 | 99.83 | 93.47 | 85.23 | 99.80 |
| $Q = 20$ | 84.20 | 89.00 | 65.60 | 62.20 | 82.90 | 79.60 | 73.70 | 83.00 |
| $Q = 50$ | 95.80 | 97.00 | 88.70 | 74.40 | 95.20 | 86.00 | 79.70 | 96.60 |
| $Q = 80$ | 98.80 | 98.90 | 96.00 | 79.50 | 97.90 | 88.80 | 79.40 | 99.80 |
| Average | 92.93 | 94.97 | 83.43 | 72.03 | 92.00 | 84.80 | 77.60 | 93.13 |

# B   More transforms

Conceptually, transform-triggered attack formulation applies to any differentiable and deterministic image transforms, as discussed in the main paper section 3. Here, we extend our adversarial formulation to two additional geometric transforms as examples — flipping and perspective transforms, to further demonstrate its flexibility.

## B.1   Flip-triggered attack

In our examination of flip-triggered attacks, we explore three specific targeted scenarios designed to activate upon the network's receipt of images subjected to vertical flip, horizontal flip, and no flip (retaining their original orientation) as inputs. The findings, as detailed in Tab. 10, reveal a notable efficacy of these flip-triggered attacks, achieving an ASR of 90% across the majority of evaluated models.

Table 10: Flip-triggered targeted attack success rate (ASR). ASRs are reported at each target flip version of image and average over all three flipping methods. Higher value indicates better attack performance. The perturbation budget is $\varepsilon = 8$. Below is the model classification accuracy (ACC) evaluation over flipped clean images without perturbation.

| Flip method | Classifier model | | | | | | | |
|---|---|---|---|---|---|---|---|---|
| | VGG19 | ResNet50 | Dense121 | Incv3 | Mobv2 | ViT-L-16 | ViT-L-32 | Swin-T |
| | ASR (%) ↑ | | | | | | | |
| None | 100.0 | 99.70 | 99.90 | 92.10 | 100.0 | 93.40 | 85.50 | 100.0 |
| Horizontal | 100.0 | 99.90 | 99.80 | 91.70 | 100.0 | 92.80 | 87.20 | 100.0 |
| Vertical | 100.0 | 99.80 | 99.90 | 97.70 | 100.0 | 96.10 | 91.40 | 100.0 |
| Average | 100.0 | 99.80 | 99.87 | 93.83 | 100.0 | 94.10 | 88.03 | 100.0 |
| | ACC (%) ↑ | | | | | | | |
| None | 100.0 | 100.0 | 100.0 | 100.0 | 87.60 | 100.0 | 100.0 | 96.00 |
| Horizontal | 93.20 | 94.20 | 96.20 | 80.60 | 87.90 | 98.30 | 96.40 | 96.10 |
| Vertical | 55.00 | 57.40 | 62.00 | 39.30 | 53.10 | 72.80 | 54.80 | 78.00 |
| Average | 82.73 | 83.87 | 86.07 | 73.30 | 76.20 | 90.37 | 83.73 | 90.03 |

Table 11: Perspective-triggered targeted attack success rate (ASR). ASRs are reported for each target perspective and averaged over all three perspectives. Higher value indicates better attack performance. The perturbation budget is $\varepsilon = 8$. Below is the model classification accuracy (ACC) evaluation over perspective transformed clean images without perturbation.

| Perspective | Classifier model | | | | | | | |
|---|---|---|---|---|---|---|---|---|
| | VGG19 | ResNet50 | Dense121 | Incv3 | Mobv2 | ViT-L-16 | ViT-L-32 | Swin-T |
| | ASR (%) ↑ | | | | | | | |
| 1 | 100.0 | 99.70 | 100.0 | 92.90 | 100.0 | 96.70 | 91.30 | 100.0 |
| 2 | 99.80 | 99.20 | 99.40 | 87.70 | 100.0 | 83.30 | 71.30 | 99.90 |
| 3 | 99.90 | 99.50 | 99.10 | 92.50 | 100.0 | 83.80 | 69.20 | 100.0 |
| Average | 99.90 | 99.47 | 99.50 | 91.03 | 100.0 | 87.93 | 77.27 | 99.97 |
| | ACC (%) ↑ | | | | | | | |
| 1 | 90.80 | 93.20 | 94.50 | 79.90 | 87.90 | 92.70 | 89.90 | 96.10 |
| 2 | 72.10 | 72.10 | 73.50 | 53.00 | 65.30 | 87.80 | 79.30 | 91.30 |
| 3 | 73.60 | 75.40 | 74.10 | 57.90 | 63.50 | 87.50 | 78.20 | 92.10 |
| Average | 78.83 | 80.23 | 80.70 | 63.60 | 72.23 | 89.33 | 82.47 | 93.17 |

## B.2  View perspective-triggered attack

Here, we introduce perspective-triggered attacks to mimic the variability encountered when taking photos from different angles. We categorize these variations into three predefined perspectives: viewing the subject from the front, from above, and from below, labeled as perspectives 1, 2, and 3, respectively. Specifically, perspective 1 maintains the image in its original state, illustrating a front-facing viewpoint. Perspective 2 simulates a downward view by transforming the image axis from $\{(0,0), (223,0)\}$ to $\{(56,56), (168,56)\}$, and perspective 3 simulates an upward view by altering the image axis from $\{(0,223), (223,223)\}$ to $\{(56,168), (168,168)\}$.

The results, as noted in Tab. 11, demonstrate the effectiveness of these perspective-triggered attacks, with targeted strategies achieving an overall ASR of over 90% when images are presented from these varied perspectives.

## C  Accuracy evaluations

In section 4.2 of our main paper, we presented targeted attacks designed to exploit vulnerabilities specific to scaling, blurring, gamma correction, and JPEG compression. To distinguish the adversarial effects from mere consequences of image transforms, we evaluate Attack Success Rate (ASR) across a diverse set of models: {VGG-19-BN, ResNet50, DenseNet-121, InceptionV3, ViT-L-16, ViT-L-32, Swin-T}.

To further investigate model sensitivity to transformations, Tab. 12 reports classification accuracy on clean images subjected to the same transformations used in attack generation. While most models maintain high accuracy (e.g., over 80%), one architecture, InceptionV3, exhibits notable sensitivity, with accuracy dropping to 69.70%. We mitigate this influence in our experiment by experimenting on sufficiently diverse set of models, and image transformations, and dataset contains larger number of instances, following similar principle to prior adversarial works that utilize image transformation for adversarial examples generation discussed in section 2.2.

## D  Memory and computation resources

We used a single NVIDIA RTX 2080Ti (12 GB) for all the experiments. Average times for generating $\{3, 5, 10\}$ target attacks are $\{2.61, 4.29, 8.67\}$ sec/image, with EOT over 3 samples for each $\bar{\theta}_i$ in equation 3. The cost scales roughly linearly with $N$ since each iteration computes one forward-backward pass per transform-target pair. Within each pair, EOT sampling (drawing multiple $\theta$ samples from $\mathcal{U}_r(\bar{\theta}_i)$) adds a constant factor per pair. In practice, we found that modest neighborhood sizes ($r = 0.1$) with a few samples suffice for robust

Table 12: Clean accuracy (ACC) evaluation over selected classification models. Higher value indicating lower classification error introduced by the image transformation.

| Transform parameter | Classifier model | | | | | | | |
|---|---|---|---|---|---|---|---|---|
| | VGG19 | ResNet50 | Dense121 | Incv3 | Mobv2 | ViT-L-16 | ViT-L-32 | Swin-T |
| $S = 0.5$ | 66.00 | 69.20 | 62.20 | 29.30 | 56.10 | 89.10 | 83.00 | 78.30 |
| $S = 1.0$ | 100.0 | 100.0 | 100.0 | 100.0 | 87.90 | 100.0 | 100.0 | 96.10 |
| $S = 1.5$ | 87.80 | 90.40 | 92.00 | 79.80 | 83.60 | 97.80 | 96.40 | 92.50 |
| Average | 84.60 | 86.53 | 84.73 | 69.70 | 75.87 | 95.63 | 93.13 | 88.97 |
| $\sigma = 0.5$ | 94.20 | 95.80 | 96.90 | 91.50 | 88.10 | 98.30 | 96.90 | 94.70 |
| $\sigma = 1.5$ | 71.00 | 76.50 | 78.30 | 67.00 | 60.30 | 88.20 | 81.90 | 81.10 |
| $\sigma = 3.0$ | 64.60 | 74.10 | 75.80 | 61.10 | 58.50 | 85.90 | 78.40 | 79.10 |
| Average | 76.60 | 82.13 | 83.67 | 73.20 | 68.97 | 90.80 | 85.73 | 84.97 |
| $\gamma = 0.5$ | 91.90 | 92.30 | 95.80 | 85.00 | 82.80 | 94.40 | 92.70 | 93.70 |
| $\gamma = 1.0$ | 100.0 | 100.0 | 100.0 | 100.0 | 87.90 | 100.0 | 100.0 | 96.10 |
| $\gamma = 2.0$ | 90.90 | 90.70 | 92.80 | 80.60 | 82.40 | 91.80 | 88.60 | 95.00 |
| Average | 94.27 | 94.33 | 96.20 | 88.53 | 84.37 | 95.40 | 93.77 | 94.93 |
| $Q = 20$ | 71.30 | 78.20 | 84.10 | 68.90 | 69.80 | 81.70 | 83.30 | 65.90 |
| $Q = 50$ | 82.10 | 86.10 | 89.60 | 75.50 | 77.10 | 87.50 | 88.00 | 82.30 |
| $Q = 80$ | 87.30 | 89.50 | 92.10 | 78.00 | 84.80 | 91.20 | 89.40 | 89.00 |
| Average | 80.23 | 84.60 | 88.60 | 74.13 | 77.23 | 86.80 | 86.90 | 79.07 |

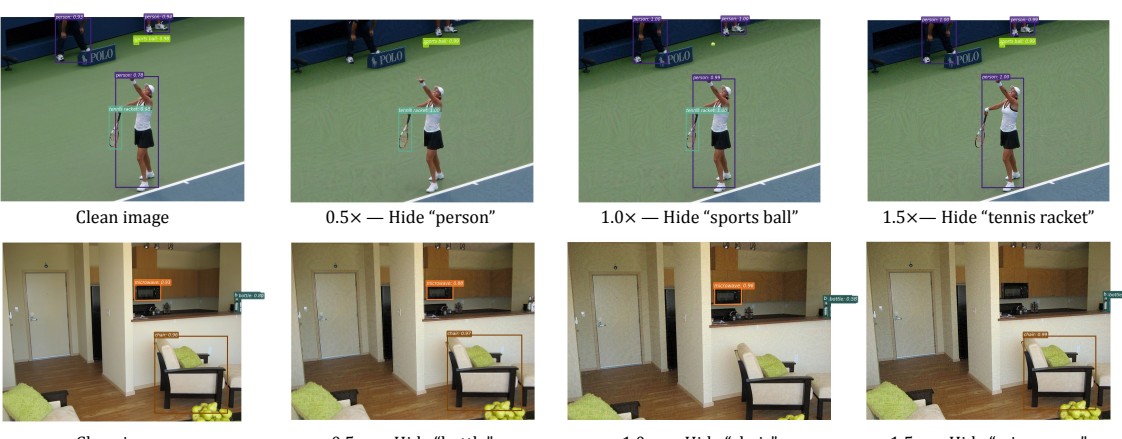

Figure 7: Visualization of scale-triggered selective hiding attack against YOLOv3. The first column shows detection results on the original clean image, while the following columns present perturbed images scaled with factors $S \in \{0.5, 1.0, 1.5\}$ (perturbation budget $\varepsilon = 10$). *Note that scaled images will have different sizes after scaling, but we present their resized versions for better display.*

trigger zones (as confirmed by the smooth loss valleys in Fig. 2), so the EOT overhead is small relative to the multi-target cost.

# E   More threat models against object detectors

Besides the transform-selective hiding threat model discussed in Sec. 4.3, here we present another **threat model: object-selective hiding.** We design a scenario where specific object classes are hidden when images are displayed at different scales, simulating real-world applications such as privacy-preserving surveillance or content-adaptive filtering. For example, certain sensitive objects (e.g., license plates or faces) could be concealed at lower resolutions in public monitoring systems, while critical details remain visible at higher resolutions for authorized analysis. In this setup, we use images containing three distinct classes and aim to hide objects from one of these classes for each transform parameter. We focus on scaling with factors

Table 13: Quantitative evaluation of scale-triggered selective hiding attack success rate (ASR) over object detection models. The higher value indicates better attack performance. Our attack formulation is generalizable to the more complex object detection task.

| Scaling factor | Detector model ASR (%) ↑ $\varepsilon = 10$ | | | | | Detector model ASR (%) ↑ $\varepsilon = 20$ | | | | |
|---|---|---|---|---|---|---|---|---|---|---|
| | Faster | YOLOv3 | FCOS | GRID | DETR | Faster | YOLOv3 | FCOS | GRID | DETR |
| $S = 0.5$ | 53.85 | 97.39 | 71.61 | 54.44 | 11.91 | 60.26 | 97.39 | 76.39 | 67.22 | 17.16 |
| $S = 1.0$ | 65.71 | 95.10 | 69.38 | 58.89 | 11.95 | 72.76 | 98.04 | 79.26 | 68.52 | 23.61 |
| $S = 1.5$ | 90.71 | 32.35 | 100.0 | 86.11 | 41.26 | 96.15 | 40.98 | 100.0 | 97.78 | 52.18 |
| Average | 70.09 | 74.95 | 80.33 | 66.48 | 21.71 | 76.39 | 78.80 | 85.22 | 77.84 | 30.98 |

$S \in \{0.5, 1.0, 1.5\}$. The attack conceals objects from class $A$ at $0.5\times$, class $B$ at $1.0\times$, and class $C$ at $1.5\times$. ASR is measured as the ratio of successfully hidden objects in the detections for each scaled input.

Tab. 13 shows that on most of R-CNN-based detectors, the scale-triggered attacks are successfully triggered when adversarial examples scale to the predefined image size, with average ASR over 66%. In Fig. 7, we showcase successful examples of object-selective hiding. The scale-triggered perturbations effectively obscure the targeted class when the perturbed image is resized to the predefined scaling factors used in transform-aware optimization, demonstrating precise control over object concealment.

## F More visual examples

For enhanced qualitative evaluation, we offer additional visual examples showcasing successful transform-triggered adversarial examples against image classification models in Fig. 8 and Fig. 9. Moreover, we present further examples of object-selective hiding attacks against object detection models in Fig. 10, Fig. 11, and Fig. 12. For transform-selective hiding attacks against detectors, which we demonstrate as a defense mechanism against image enhancement (Fig. 3 in the main text). In the supplementary material, we include animations in `.gif` format to illustrate how object detectability changes dynamically under different enhancement transformations.

**Attacks against classifiers.** The examples in Fig. 8 and Fig. 9 illustrate that with imperceptible noise perturbation, an image can be misclassified as multiple target labels when subjected to specific image transformations.

**Attacks against detectors.** In the scenario of object-selective hiding attacks, consider the images in the first row of Fig. 10 as an illustration. It demonstrates that in the clean image, objects labeled as three distinct categories (*person, ski,* and *snowboard*) are detected. However, upon adding scale-triggered perturbations, objects labeled as one of these categories become hidden in three differently scaled versions of perturbed images, as depicted in the titles: $0.5\times$ — *Hide "person"*, $1.0\times$ — *Hide "ski"*, and $1.5\times$ — *Hide "snowboard"*.

In the scenario of transform-selective hiding attacks, we simulate image enhancement processes—zoom-in, deblur, and gamma correction—applied to raw images initially presented as distant scenes, blurry images, or low-light conditions. The animations included in the submission illustrate the effectiveness of our transform-triggered examples: objects remain detectable in unaltered images but become concealed when the images undergo enhancement. This demonstrates the perturbation's ability to selectively obscure objects under specific transformations while preserving detectability close to the original, non-perturbed state.

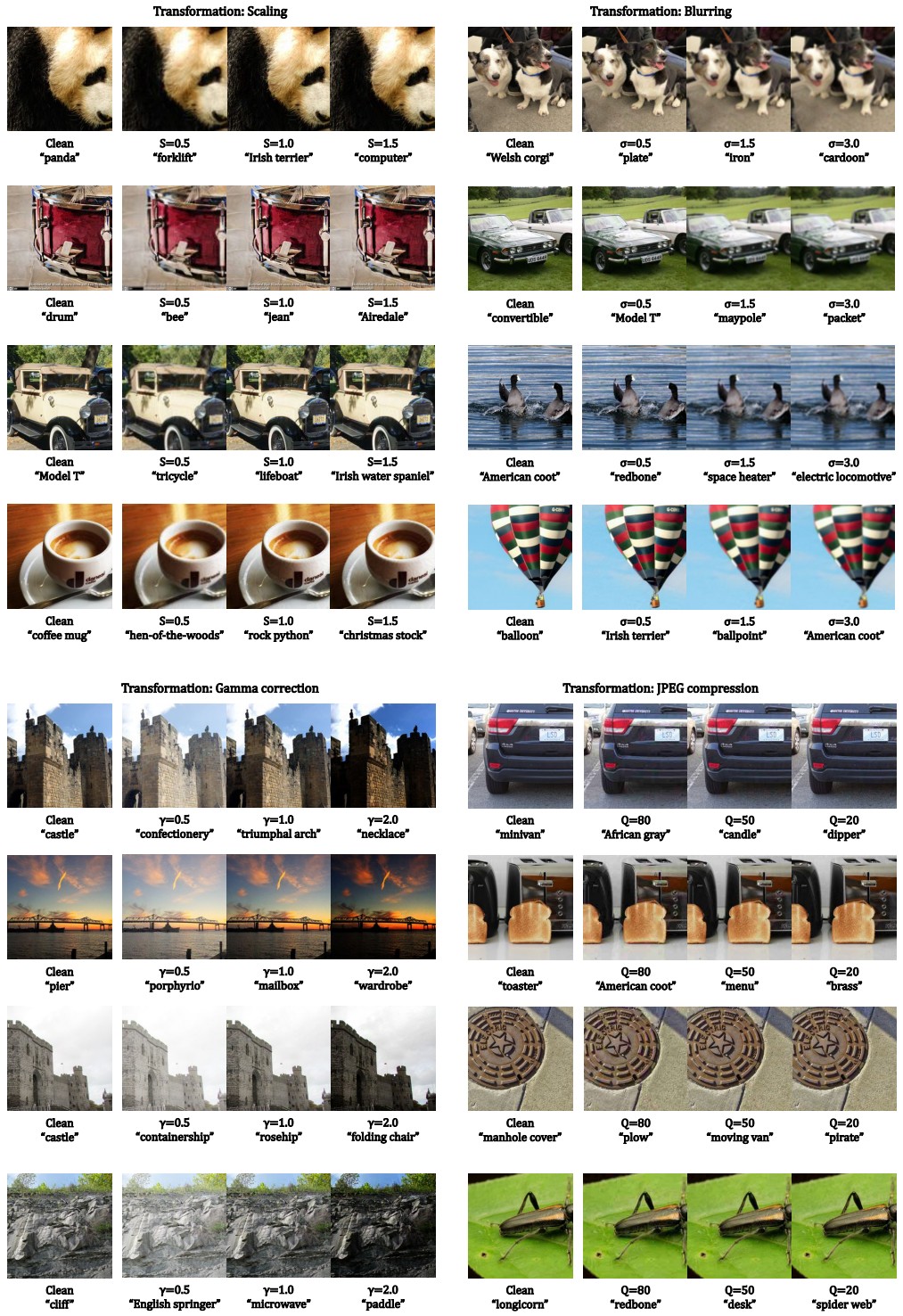

Figure 8: Visual examples for transform-dependent attacks against image classifiers. In this figure, we show visual effects of clean image and the perturbed images transformed with different parameters.

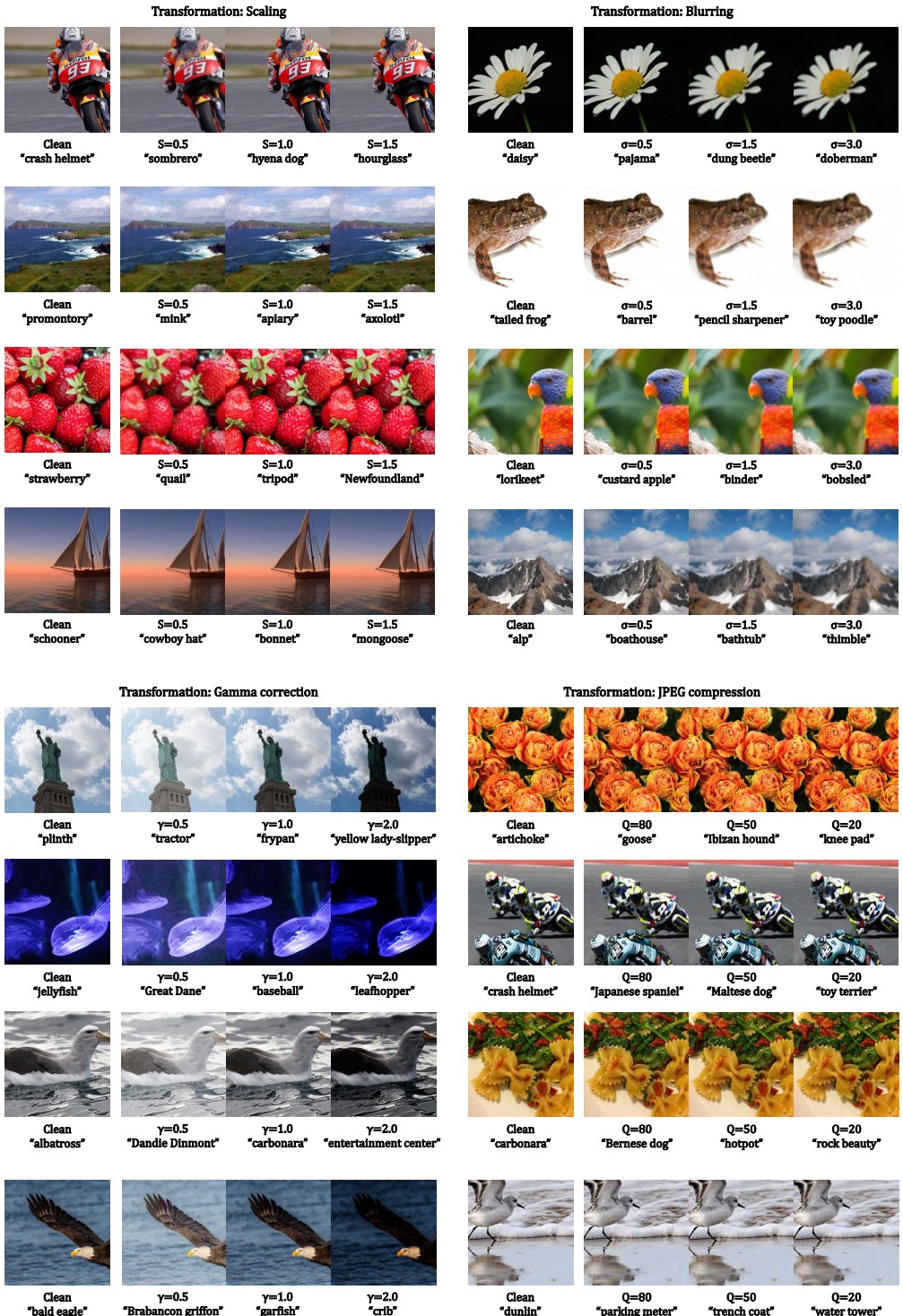

Figure 9: Visual examples for transform-dependent attacks against image classifiers. In this figure, we show visual effects of successful attacks under different image transformations with different transform parameters.

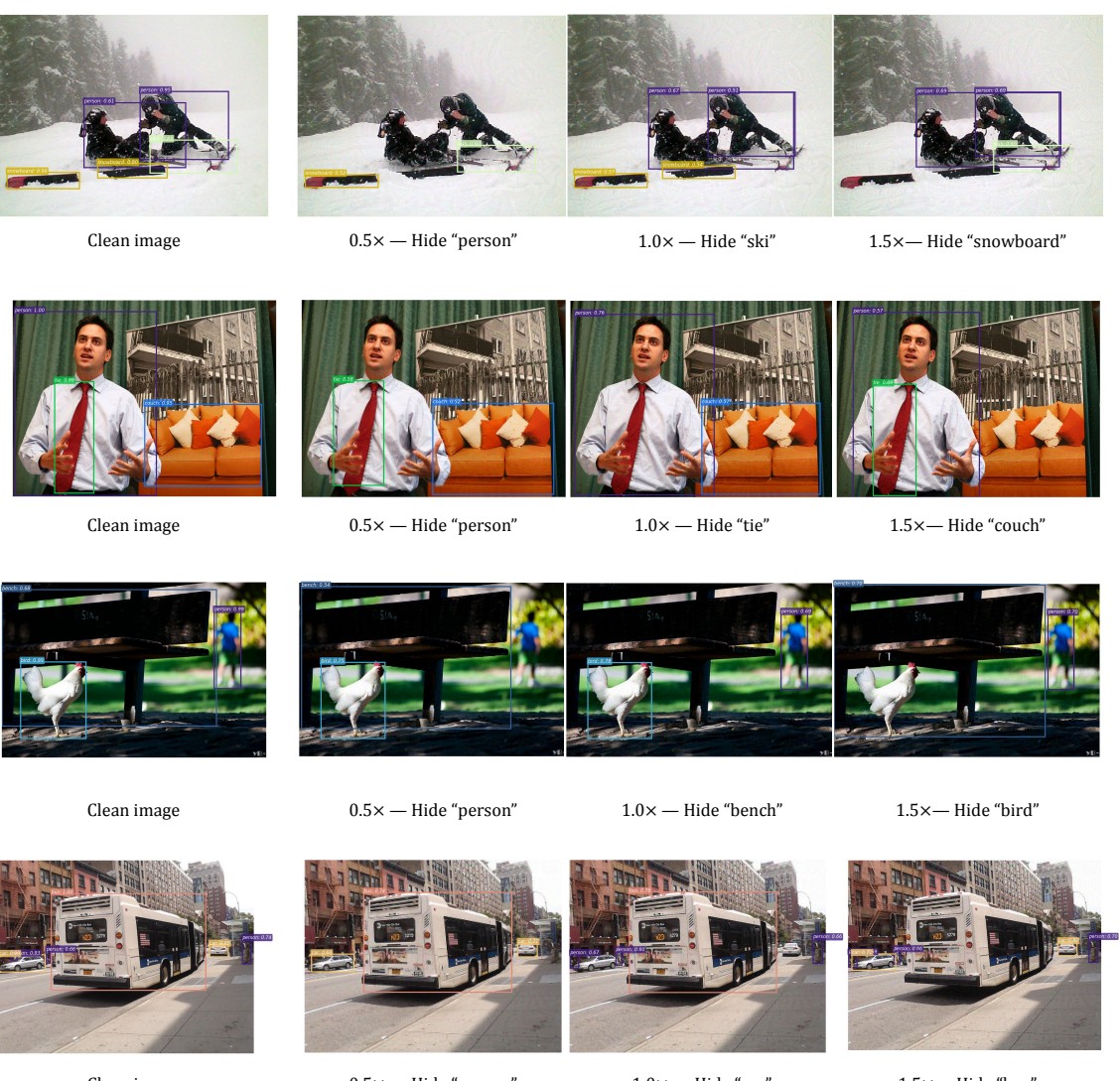

Figure 10: Visual examples of scale-dependent selective hiding attacks against object detection model `FCOS`. From top to bottom, `ImageIDs`: `000000142790`, `000000170099`, `000000197870`, `000000338625`. Note, images labeled as $0.5, 1.5\times$ are in resolutions different from the original image, and they are resized to the same size for better display.

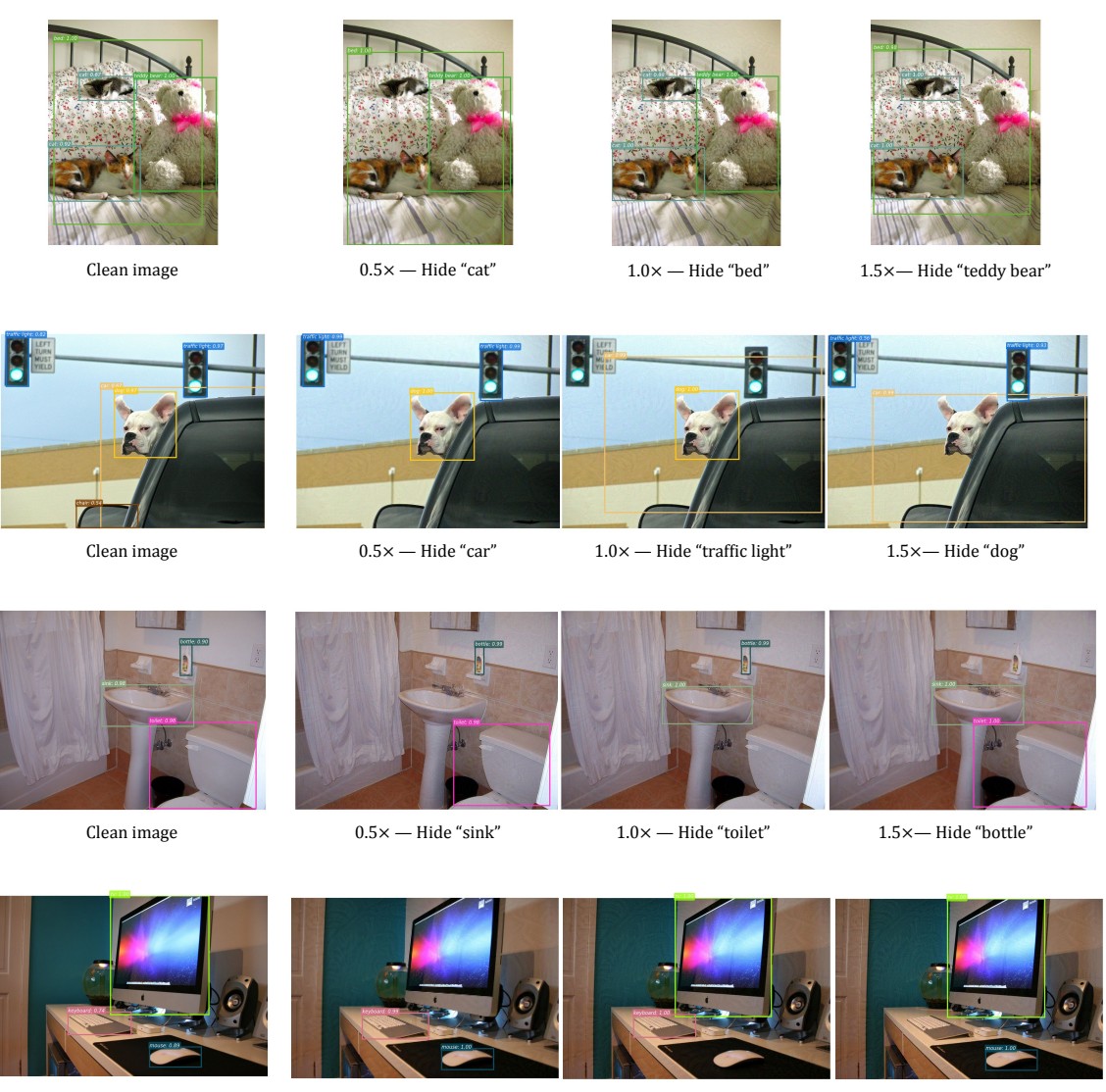

Figure 11: Visual examples of scale-dependent selective hiding attacks against object detection model `YOLOv3`. From top to bottom, `ImageIDs: 000000478393, 000000076417, 000000167898, 000000186282`. Note, images labeled as $0.5, 1.5\times$ are in resolutions different from the original image, and they are resized to the same size for better display.

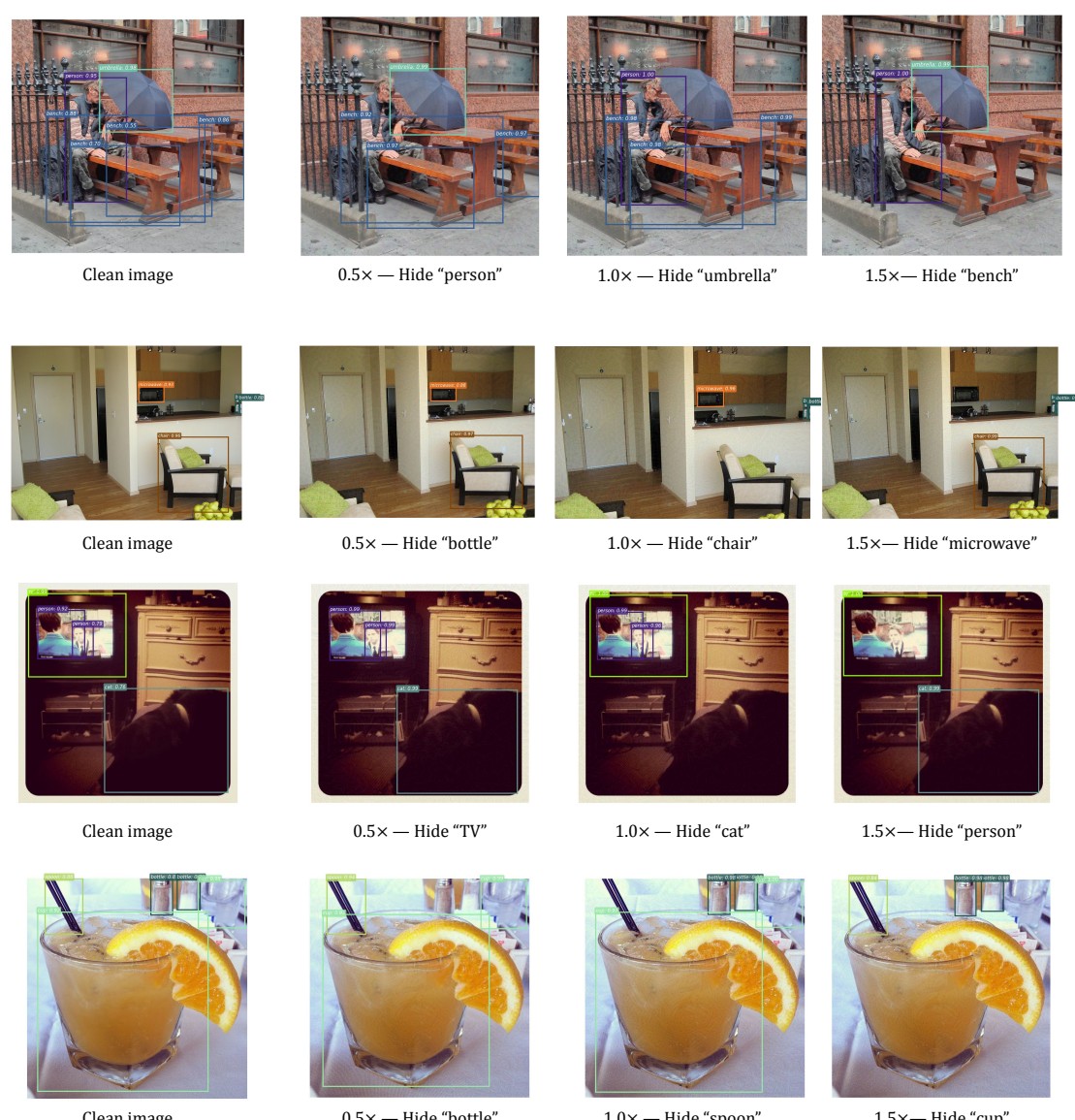

Figure 12: Visual examples of scale-dependent selective hiding attacks against object detection model `Faster R-CNN`. From top to bottom, `ImageIDs: 000000455157, 000000488075, 000000169076, 000000463283`. Note, images labeled as $0.5, 1.5\times$ are in resolutions different from the original image, and they are resized to the same size for better display.

