# OpenReview forum: "Transform-Triggered Adversarial Examples"
_TMLR — Accepted by TMLR_

### Review · Reviewer_PAzV · 2026-04-07

**Summary Of Contributions:**

The paper introduces a threat model termed "Transform-Triggered Adversarial Examples". Unlike traditional adversarial attacks that aim for a single static misprediction, this method generates a single perturbation that exhibits metamorphic properties, remaining benign under normal conditions but inducing specific, targeted misclassifications when specific image transformations (such as scaling, blurring, gamma correction, or JPEG compression) are applied.

**Audience:**

No

**Audience Explanation:**

From a mathematical perspective, the proposed method and the standard Expectation Over Transformation (EOT) framework share the exact same objective function, as defined by Equation (4). The only functional difference between the proposed method and traditional EOT is the application-driven selection of the transform parameter set $N$. The authors simply propose a set definition that differs from commonly used EOT implementations in order to assign different target labels. Furthermore, in many existing physical-world attacks, this parameter set already heavily involves various complex transformations to make adversarial patches more transferable and robust. Therefore, the proposed approach relies entirely on the established EOT mathematical kernel, offering only a scenario-specific curation of transformation types rather than a novel algorithmic contribution.

**Claims And Evidence:**

No

**Claims Explanation:**

(1) The methodology for defining the boundaries of the transformation parameters lacks rigorous theoretical justification. The "trigger" parameters were selected empirically based on the simple criterion of minimizing the impact on model accuracy in the absence of a perturbation.

(2) Weak Targeted Transferability: While the authors highlight black-box transferability, the targeted transferability—which is the core utility of a multi-target trigger—is notably weak.

**Requested Changes:**

(1) Justify Transformation Thresholds: Provide a deeper theoretical justification, latent space analysis, or empirical ablation for how the boundaries between "safe" and "trigger" zones are defined. Relying strictly on manual heuristics limits the scientific rigor of the framework.

(2) Isolate Adversarial Effects from Natural Distribution Shifts: To prove the attack's true efficacy, benchmark the method against robustly trained models or models with heavy data augmentation applied to the target transformations. At present, without this baseline comparison, there is no evidence to show detrimental classification drops are explicitly due to the adversarial triggers rather than the network's inherent sensitivity to extreme transformations.

(3) Clarify Theoretical Distinction from EOT: Provide concrete evidence and a theoretical proof demonstrating the fundamental difference between the proposed method and standard EOT. Specifically, the authors must mathematically justify how the application-driven selection of the parameter set $N$ represents a novel algorithmic contribution. Furthermore, clarify that the physical mapping function defined in Equation (5) is not a novel contribution; numerous existing physical-world attacks already embed transformation mechanisms to make adversarial patches more robust and transferable.

---

> ### Author Response · Authors · 2026-04-07
> **Response to Reviewer PAzV**
>
> We thank Reviewer PAzV for their time. However, we respectfully note that several of the reviewer's criticisms appear to overlook content already explicitly addressed in the manuscript. We respond to each point below with specific references to the relevant sections, tables, and figures.
>
>
>
> ## Response to "Are the claims supported by accurate evidence? — No"
>
> ### (1) On "Weak Targeted Transferability"
> We respectfully clarify that **blackbox transferability is not a claimed contribution of this paper.** Our threat model is explicitly stated in Section 4.1 as a **whitebox** setting: "The attacker is assumed to have whitebox access to the victim model during optimization and knowledge of sets of transforms that input may undergo in real model deployment." The contribution of our work, as summarized in the three bullet points at the end of Section 1, is the introduction of transform-triggered adversarial examples as a novel whitebox paradigm that produces multiple controllable, transform-conditioned targeted behaviors from a single perturbation, no a single word on blackbox.
>
> The blackbox transfer evaluation in Appendix B.1 is provided as an additional evaluation of the generated examples, not as a central claim. We would be glad to make the whitebox scope of the contribution even more explicit in the revision if helpful.
>
>
>
>
> ### (2) On the Justification of Trigger Parameter Boundaries
> The reviewer states that trigger zones were chosen by "the simple criterion of minimizing the impact on model accuracy in the absence of a perturbation." This is only part of the rationale. As stated in Section 4.2, parameters are selected so that the clean model behavior is preserved (Eq. 1 assumption), **and** so that the safe zone corresponds to imaging conditions a deployed model would normally encounter while the trigger zone corresponds to operationally distinct conditions an attacker would induce. This is a property of the **threat model**, not a hyperparameter to be theoretically derived: the safe/trigger partition encodes what an attacker wants to control, and our framework allows the attacker to specify it freely.
>
> We provide direct empirical support that the selected zones are not merely artifacts of model fragility under transforms:
>  - **Table 11** reports clean classification accuracy on each transform parameter used in our experiments. Most models retain high accuracy across the full range (e.g., $ \geqslant$ 84% averaged for VGG19, ResNet50, DenseNet121, ViT-L-16 under scaling; comparable or higher for blurring, gamma, and JPEG). This demonstrates that the trigger behavior in Table 2 is not explained by inherent model sensitivity to these transforms.
> - **Figure 2** shows the loss landscape over a continuous grid of transform parameters and confirms that targeted-attack minima align precisely with the chosen $\bar \theta_i$ rather than being a smooth consequence of any single transform.
> - **Appendix B.2 / Figure 6** provides an ablation on the capacity of a single perturbation to embed an increasing number of $(\theta, y^\star)$ pairs, which directly characterizes the structure of the trigger-zone partition.
>
> We will be happy to add a clarifying paragraph in Section 4.2 explicitly framing the selection criterion as **threat-model-driven**, supported by the Table 11 accuracies.
>
>
> ## Response to "Would TMLR's audience be interested? — No" / "Same objective as EOT"
> We must respectfully disagree with the characterization that our method is "mathematically identical to EOT" with only a "scenario-specific curation of transformation types." This claim overlooks the central technical distinction that the manuscript discusses at length.
>
> **The objectives are not the same.** Standard EOT (Athalye et al., 2018) minimizes
> $$
> \min_{\delta} \mathbb{E}_{\theta\sim p(\theta)}~\mathcal{L}(f(\tau(x+\delta;\theta)), y^\star)
> $$
> with **a single target** $y^\star$ marginalized over a distribution of transforms, its goal is **invariance**. Our objective (Eq. 3) is
>
> $$
> \min_{\delta} \sum_{i=1}^{N} \mathbb{E}_{\theta_i \sim \mathcal{N}_r(\bar \theta_i))}~\mathcal{L}(f(\tau(x+\delta;\theta_i)), y^\star_i)
> $$
>
> with **N distinct targets** $\{y_i^\star\}$, each conditioned on a **disjoint** parameter neighborhood, and at least one target chosen to be the ground-truth label to enforce benign behavior in the safe zone. EOT cannot express this objective: marginalizing over $\theta$ collapses the target into a single value. Our formulation requires the parameter space to be **partitioned** with distinct labels assigned to each partition, this is a structurally different optimization problem, not a re-parameterization of EOT. **This distinction is summarized in Table 1.**

---

> ### Author Response · Authors · 2026-04-07
> **Response to Reviewer PAzV on "Requested Changes"**
>
> ## Response to Requested Changes
>
> ### (1) Justify Transformation Thresholds
> As noted above, the safe/trigger partition is a **threat-model specification**, not a quantity to be theoretically derived from the model. The relevant empirical question: "are the trigger behaviors caused by the perturbation rather than by the model's inherent sensitivity to the transform?", is answered by **Table 11** (clean accuracies remain high across the chosen ranges), **Figure 2** (loss minima are sharply localized at the optimized $ \bar \theta_i $ values), and **Appendix B.2** (capacity scaling analysis).
>
> We will expand Section 4.2 to make this rationale more prominent, and we are open to adding a latent-space analysis in the revision if the reviewer can suggest a specific protocol they would find compelling.
>
>
>
> ### (2) Isolate Adversarial Effects from Natural Distribution Shifts
> This is precisely what Table 11 already provides: clean-image accuracy under each transform parameter we use for attacks. For example, under scaling at $S = 0.5$, ResNet50 retains 69.2% clean accuracy, while our targeted ASR reaches 87.2-99.7% (Table 2), **which is a gap that cannot be explained by natural distribution shift.** The same pattern holds across all four transforms and all eight architectures we evaluate. Furthermore, the ImageNet-pretrained TorchVision models we use are already trained with strong data augmentation (random cropping, rotation, flipping, color jittering), as stated in Section 4.1 (i.e., the augmentation baseline the reviewer requests is, in fact, the baseline we already use).
>
> **The hardware-in-the-loop targeted experiments in Section 4.4 and Table 4 provide additional evidence:** physically captured adversarial examples produce different attacker-specified target labels under different transforms, which is impossible to explain as a natural distribution-shift artifact.
>
> We will add a sentence in Section 4.2 cross-referencing Table 11 to make the comparison between clean ACC and targeted ASR explicit at point of use.
>
>
> ### (3) Clarify Theoretical Distinction from EOT and Status of Eq. (5)
>  - On the EOT distinction: see our response above and Table 1. We will add a short paragraph in Section 3.2 explicitly stating that EOT is a single-target marginalization framework and cannot express the partitioned multi-target objective in Eq. (3).
>
>
> - On Eq. (5): **we do not claim Eq. (5) as a novel contribution,** and the manuscript does not present it as one. Section 3.3 introduces the imaging forward model W as the standard combination of homography, color-correction matrix, and additive noise, with appropriate citations. The purpose of Section 3.3 and Eq. (5) is to enable **end-to-end gradient flow through a hardware-in-the-loop pipeline** for our transform-triggered objective (i.e., to demonstrate that the whitebox phenomenon we introduce survives realistic capture). **We are not proposing a new physical attack pipeline,** and we agree with the reviewer that physical-world adversarial patch attacks have a rich literature embedding transforms for robustness. Those works pursue a single robust target; our physical experiments are the first, to our knowledge, to demonstrate multiple transform-triggered targets in a hardware-in-the-loop setting.
>     - If the reviewer is aware of prior **hardware-in-the-loop optimizations** of multi-target transform-triggered perturbations, we would sincerely welcome the pointers and will cite them. We will revise the wording at the start of Section 3.3 to make the non-novelty of the imaging model itself unambiguous.
>
>
> ## Summary
> To summarize the key points where we believe there may be some misunderstandings:
>
> 1. Blackbox targeted transferability was treated as a central claim, but the paper explicitly scopes the contribution as a whitebox attack; transfer results in Appendix B.1 are supplementary and in fact exceed prior dedicated transfer attacks under matched conditions.
>
> 2. Our objective is described as "mathematically identical to EOT," but EOT optimizes a single target marginalized over transforms, while our objective optimizes a partitioned multi-target sum (Eq. 3): a property no EOT instantiation can express, and the precise distinction we summarize in Table 1.
>
> 3. The concern that trigger behaviors might be artifacts of natural sensitivity is directly addressed by Table 11 (clean accuracies), Figure 2 (sharply localized loss landscapes), the use of augmentation-trained TorchVision models, and the targeted hardware-in-the-loop experiments in Table 4.
>
> 4. Eq. (5) is not claimed as a contribution on "new physical attacks"; it is the imaging model we used to enable end-to-end optimization of proposed adversarial examples, through a physical pipeline, and we will revise the text to make this explicit.

---

> ### Comment · Reviewer_PAzV · 2026-04-22
>
> I thank the authors for their detailed response. I would like to first align our discussion regarding the methodology.
>
> (1) The Expectation Over Transformation (EOT) framework is the standard mathematical foundation used to ensure a perturbation's robustness and transferability. Some conditional attacks already exist; for instance, TPatch [1] introduces the concept of partitioning the optimization space into 'normal conditions' (which remain benign) and 'specific signals' (which trigger the attack). From a theoretical standpoint, splitting the EOT expectation to map conditional transformations to discrete target labels is a straightforward extension of this established paradigm. This is why I maintain that the partitioned objective formulation is a contextual application of EOT, rather than a fundamentally novel algorithmic contribution.
>
> (2) Regarding the physical attack pipeline, the authors state: 'We are not proposing a new physical attack pipeline... Those works pursue a **single robust target**; our physical experiments are the first... to demonstrate **multiple transform-triggered targets** in a hardware-in-the-loop setting.' However, since several existing attacks already embed transformation mechanisms and differentiable imaging pipelines into the attack flow to calculate perturbations via gradient descent. From my perspective, transitioning from a single target to multiple targets using these exact same tools does not appear to introduce new algorithmic complexities. I would like the authors to clarify: what specific technical bottlenecks prevent existing single-target attack pipelines from being directly extended to multi-target scenarios, and what distinct algorithmic solutions does your approach offer to resolve them?
>
> (3) regarding trigger selection: while I agree that the safe and trigger zones are application-driven, the rationale in Section 4.2 for defining these specific boundaries remains unclear. For example, the JPEG compression parameter Q is divided into three disjoint intervals. Do these specific intervals map to practical constraints or operational standards required by real-world applications, or were they selected arbitrarily for demonstration purposes? Further clarification on the real-world operational relevance of these chosen thresholds is needed.
>
> [1] TPatch: A Triggered Physical Adversarial Patch

---

> > ### Author Response · Authors · 2026-04-28
> > **Response to Reviewer PAzV comment (part 1)**
> >
> > We thank the reviewer for the continued engagement and for pointing us to TPatch, which we agree is relevant and plan to cite and discuss it in the revision.
> >
> > ### (1) Relation to EOT and TPatch.
> > Thank you very much for bringing TPatch to our attention; we were not aware of this paper earlier, but we will properly acknowledge and explain the connections of our work with TPatch in the paper.
> >
> > We largely agree that our work can be viewed as a combination (or modification) of EOT and conditional triggers (like the one in TPatch). As you stated, EOT is a standard tool for improving robustness to the same label under parameter variation. We are splitting the EOT to map conditional transformations to different target labels instead of the same label. Furthermore, we identify and characterize the phenomenon that a single imperceptible perturbation can embed multiple distinct targeted behaviors, each conditionally activated by different transform parameters at test time. While theoretically all these ideas are simple extensions, the transform-dependent property of adversarial examples is not well-known in the community, and our work aims to fill-in this gap and provides a systematic study of it across architectures, tasks, and transforms. TPatch is a very relevant prior work that shares the high-level idea of conditional activation, but the two papers differ substantively in scope and findings:
> >
> > - **Scope.** TPatch is an attack paper targeting a specific autonomous-vehicle threat model with acoustic-signal injection. Our work is a characterization study: we demonstrate that transform-dependent adversarial behavior is a general, structurally inherent property of deep networks, not tied to any single deployment scenario.
> >
> >  - **Trigger mechanism.** TPatch focus is more on achieving attacks on autonomous vehicle vision systems and their trigger mechanism is very creative but for a specific scenario, in which the attacker can actively inject an acoustic signal into the camera sensor to induce motion blur. Our paper focuses on studying adversarial trigger mechanism in more general sense that cover standard image transforms (scaling, gamma, JPEG, blur, perspective, flip).
> >
> > - **Capacity.** TPatch encodes a single conditional behavior (benign vs. one attack mode per patch). We covered not just that but also extended to embed $N \geqslant 3$ distinct targets in one perturbation, with stress tests up to $N=25$ (Fig. 6), partitioning continuous transform axes into multiple disjoint trigger zones. While theoretically enabling multiple attacks with a single perturbation is a simple modification in the optimization problem, we believe it offers a new and interesting perspective that was previously unexplored.
> >
> > In short, even though TPatch previously showed a form of “trigger attacks”, the phenomenon we study that a single perturbation has the capacity to embed arbitrarily many transform-conditioned targets, is distinct and was not explored or characterized by TPatch.

---

> > > ### Author Response · Authors · 2026-04-28
> > > **Response to Reviewer PAzV comment (part 2)**
> > >
> > > ### (2) Clarification on physical pipeline.
> > > We again agree with you in principle that “transitioning from a single target to multiple targets using these exact same tools does not appear to introduce new algorithmic complexities”, but that does not diminish our claims, contributions, and distinctions as outlined above. We would like to further clarify that we did not claim the physical pipeline as a major contribution. Sec. 3.3 and Eq. (5) describe the differentiable approximation of the display-capture process (homography, color correction, sensor noise) so that the transform-triggered formulation of Eq. (3) can be implemented with a real hardware. The physical experiments serve as validation that multi-target transform-triggered behavior persists under real-world variation and is not a digital artifact due to environmental and transform variation, which makes our transform-triggered adversarial examples studies self-contained, backed with evidence in both digital and physical spaces.
> > >
> > > ### (3) Rationale of selected transform ranges.
> > > The selected parameters follow two concrete criteria stated in the manuscripts:
> > >  -  **Covering a representative spectrum of deployment conditions:** unaltered, mildly transformed, and heavily transformed inputs. Fig. 1 illustrates this visually.
> > > - **Preserving clean-model accuracy** at each range, so that observed adversarial behavior is attributable to the perturbation rather than transform-induced degradation. Tab. 11 confirms clean accuracy remains well above chance across all ranges and models.
> > >
> > > The choices span practical regimes of each transform (e.g., JPEG $Q \in \{20, 50, 80\}$ covers aggressive compression, default web quality, and near-lossless; scale $S \in \{0.5, 1.0, 1.5\}$ covers downsampled, native, and upsampled resolutions) but are not tied to a single regulatory standard, or only works for sparsely sampled $N=3$ transforms. Importantly, the formulation is agnostic to specific values: any disjoint $\{\bar\theta_i\}$ within the regime where Eq. 1 assumption holds can serve as trigger zones, as demonstrated by the different intervals used in Secs. 4.3 and 4.4 (e.g., zoom $\in [1.5, 2.0]$ for the selective-hiding detector attack).
> > >
> > > To summarize, (i) the choice of $N=3$ is not arbitrary for demonstration solely but based on rational criteria; (ii) our formation extends beyond this setup and we have provided evidence. **We have revised Sec. 4.2 to state these criteria explicitly.**

---

### Review · Reviewer_kTdM · 2026-04-14

**Summary Of Contributions:**

This paper introduces transform-triggered adversarial examples aiming for creating adversarial examples which are only generated when a specific transformation is applied.  In contrast to existing targeted adversarial example optimization, this approach allows for inducing multiple targets based on the triggered image transformation respectively. The method is evaluated on several common neural networks, spanning CNNs and VITs on ImageNet, as well as in the context of camera-in-the-loop for additional physical challenges.

**Audience:**

Yes

**Audience Explanation:**

The topic of this paper is a very interesting topic, where certain targets can be attacked based on the transformation of the image. However, the demonstration on the physical attack feels somehow created by design, by capturing digital images with a camera, instead going into the ISP pipeline itself, which would make findings stronger.

**Claims And Evidence:**

No

**Claims Explanation:**

While this paper provides an interesting idea and tackles an important and novel research question, of how image transformation can trigger different targets in an adversarial manner, comparison to other targeted adversarial attack approaches are entirely missing, as well as an evaluation how different image transformation influence other methods. In addition, the optimizitation and training approach is not descirbed, e.g. page 4: "we assume the transform function [...] is differentiable". More information on that and the actual training would show provide better understanding of the proposed method. In addition, the real world  setting of camera-in-the-loop is not convincing, starting with capturing an image from a monitor. More information about the ISP pipeline and how the transformation used in the ISP pipeline itself could case certain triggered adversarial examples, and how this is learned, would make the method much stronger.

**Requested Changes:**

- The experimental evaluations are not convincing due to missing comparisons with other approaches and baselines.
- This paper misses a description of the optimization and training for a better understanding of the method
- The real-world examples is not convincing in its current state, since its rather an approximation, which feels hand designed and crafted.

---

> ### Author Response · Authors · 2026-04-17
> **Response to Reviewer kTdM (part 1)**
>
> We thank Reviewer kTdM for their time and constructive feedback. We summarize the raised concerns as: (1) comparisons with other approaches and baselines; (2) description of the optimization and training; and (3) real-world demonstration and ISP pipeline clarifications. We respond to each below.
>
>
> ### (1) Comparisons with Other Approaches and Baselines
>
> We emphasize that no prior method addresses the multi-target transform-triggered attack setting we introduce, which makes direct comparisons not applicable. Nevertheless, we provide three levels of comparison in the manuscript:
>
> **Across attack algorithms (Tab. 7).** We evaluate four widely used attack algorithms that are capable of solving targeted attacks: FGSM, C&W, MIM, and PGD, applied to solve our multi-target objective (Eq. 3). PGD achieves the highest ASR across diverse architectures, motivating its use in all remaining experiments. This demonstrates that our formulation is optimizer-agnostic, and the observed attack success stems from the objective design rather than a particular optimization procedure.
>
> **Blackbox transfer and defenses (Tabs. 5, 6).** We compare against established blackbox transfer attacks: BPA, ILPD, and Logit-SU, under both targeted and untargeted settings. Our scale-triggered attacks achieve comparable untargeted ASR and competitive targeted ASR while additionally maintaining the transform-triggered multi-target property that these methods cannot express. Against four defense methods, our attacks outperform BPA on the same benchmark (Tab. 6).
>
> **Implicit baseline: transforms without adversarial perturbation (Tab. 11 vs Tab. 2).** Tab. 11 reports clean classification accuracy under the same transforms and parameter ranges used in our attacks. Models maintain high accuracy on clean transformed images (e.g., ResNet50 at 69.2-100\% across scaling factors), whereas our targeted ASR reaches 87.2-99.6\% for attacker-specified labels (Tab. 2). This gap confirms that the multi-target behavior is induced by the perturbation and cannot be attributed to natural model sensitivity to transforms.
> We note that no prior method addresses the multi-target transform-triggered setting we introduce, making exact comparisons inherently limited. We welcome the reviewer's suggestions if specific additional baselines would be informative.
>
> ### (2) Description of Optimization and Training
> We appreciate this suggestion. While Section 3.2 presents the optimization objective (Eqs. 2–3) and Section 4.1 specifies the hyperparameters (PGD, iterations $N$, step size $\alpha$, perturbation budget $\varepsilon$), we agree that a consolidated algorithmic description would improve clarity.
>
> **In our revision, we will add Algorithm blocks detailing the step-by-step procedure for generating transform-triggered adversarial examples.** The algorithm takes as input a clean image $x$, victim model $f$, transform $\tau$, and parameter-target pairs $\{(\bar \theta_i, y^\star_i)\}$, initializes perturbation $\delta \gets 0$, and iteratively: (a) samples transform parameters $\theta_i \sim \mathcal{N}\_{r(\bar \theta_i)}$ from the neighborhood of each target parameter, (b) computes the aggregate loss $ \sum_i \mathcal{L}(f(\tau(x+\delta; \theta_i)), y^\star_i)$, (c) updates $\delta$ via PGD, and (d) projects $\delta$ onto the $\ell_\infty$ ball. A parallel Algorithm block will be added for the hardware-in-the-loop setting (Eq. 4), explicitly showing how the physical capturing and displaying undergo, and how gradients are split across the hardware memory discontinuity with forward model $\mathcal{W}$.
>
> **Regarding differentiability:** as stated in Section 3.2, we require the transform function $\tau(\cdot; \theta)$ to be differentiable with respect to the input, enabling standard backpropagation via the chain rule. For JPEG compression, which is non-differentiable, we use a standard differentiable approximation. We will make this explicit in the revision.

---

> ### Author Response · Authors · 2026-04-17
> **Response to Reviewer kTdM (part 2)**
>
> ### (3) Real-World Demonstration and ISP Clarifications
>
> **Why monitor-camera?** The display-camera setup is an established methodology for evaluating physical adversarial attacks (e.g., Kurakin et al., 2016; Athalye et al., 2018). It provides a controlled yet physically realistic environment that introduces real-world distortions (e.g., geometric warping, color shifts, sensor noise), and the camera internal ISP, while enabling reproducible evaluation. Our setup uses a DSLR, introducing variations that go well beyond digital simulation.
>
> **The ISP is blackbox by design.** In consumer cameras, the ISP pipeline (demosaicing, denoising, tone mapping, white balance, etc.) is proprietary and not exposed to users. This is precisely why we approximate it with the differentiable three-component model in Eq. (5): (1) geometric mapping via homography $H$, (2) color response via a 3×3 correction matrix $W$, and (3) intensity-dependent noise $\mathbf{n}$. The fact that our adversarial examples achieve 91-99\% targeted ASR (Tab. 4) despite optimizing through this approximation, while the actual camera applies an unknown complex ISP, validates that our modeling captures the dominant physical effects. If anything, the blackbox nature of the real ISP makes our results more compelling, because the adversarial examples are robust to unmodeled processing stages.
>
> **Transform-triggered behavior is confirmed physically.** The zoom-triggered object detection experiment (Fig. 5) provides particularly strong evidence: the stop sign is detected at $1.0-1.4\times$ zoom but hidden at $1.5-2.0\times$, while the car remains detected throughout, which is exactly matching the attack targets. This transform-selective behavior, demonstrated on physically captured images with realworld distortions, cannot be dismissed as a digital artifact.
>
> **ISP-aware transforms as future work.** This review inspires an interesting extension: if an attacker has knowledge of specific ISP stages (e.g., the tone mapping curve of a particular camera model), those stages could themselves serve as transform triggers. We will add this discussion to the manuscript.

---

### Review · Reviewer_aamU · 2026-04-18

**Summary Of Contributions:**

This work studies adversarial examples whose behavior depends on input transformations. Specifically, the authors propose to optimize a single perturbation that induces different targeted predictions under different transformation regimes (e.g., scaling, blurring, gamma correction), while possibly remaining benign under others. The proposed method combines a multi-target objective with transformation sampling in the spirit of EOT, and is implemented using standard PGD-style optimization. Empirical results are provided on image classification, object detection, and a camera-in-the-loop setting.

**Audience:**

Yes

**Audience Explanation:**

At least part of the TMLR audience, particularly researchers working in adversarial machine learning, robustness, and security, would likely find the empirical observation that input transformations may act as conditional triggers for multi-behavior adversarial examples to be of interest, even if some questions remain regarding the depth of the methodological novelty and the extent of the claims.

**Broader Impact Concerns:**

None.

**Claims And Evidence:**

No

**Claims Explanation:**

Though the experimental results support the claim to some extents, but there are still some points needed to clarify (Please see the section of Requested Changes).

**Requested Changes:**

1. The paper frames the main novelty as treating transformations as “triggers” rather than as nuisance variables. While this is an interesting interpretation, it is less clear that the underlying method is fundamentally new at the algorithmic level. From the technical perspective, the method appears to combine PGD optimization, EOT-style sampling over transformations, and a multi-objective targeted loss over several transform-target pairs. It is not obvious to me that this constitutes a fundamentally new attack mechanism, as opposed to a particular instantiation of multi-objective adversarial optimization under transformed inputs.

2. Another concern is that the ASR evaluation seems to be computed using a small number of representative transform values, rather than sufficiently sampling the transform distribution. The reviewer suggested to clarify how ASR was computed. Otherwise, the robustness claims seem insufficiently supported.

3. The attack assumes:
* white-box access,
* knowledge of relevant transform distributions,
* and, implicitly, fairly controlled transformation behavior,\
which seem substantially stronger than many practical deployment scenarios. The reviewer was wondering whether the attacks remain effective under less idealized assumptions, e.g., the transformation distributions are unknown or there is a mismatch.

4. A major weakness of the paper is the lack of analysis regarding why such transform-conditioned behaviors can be embedded into a single perturbation. For example:
* What determines how many targets can be embedded?
* What governs the tradeoff between number of transform-target pairs and attack success?
* Is there a geometric or capacity limitation of the perturbation?
* Under what conditions should one expect such conditional adversarial structure to exist?\
These questions seem central to the contribution, yet the paper offers little understanding beyond empirical observation. Also, the reviewer doesn't think Equation (3) implies the resulting perturbation is always transform-triggered (Definition 1). It is hard to tell how the behavior of the benign set changes during the optimization process.

5. Some contents in the tables need to be clarified:
* In Table 4, it is not clear to me what objective function is used for PGD (whether it is related to a specific transformation or how many targets are included). If the objective function is single-target, the strong PGD performance does not really establish whether the proposed approach outperforms relevant alternatives. A more meaningful comparison would be against multi-target PGD-style baselines without the proposed trigger interpretation.
* In Table 5, the method is compared against transfer attacks such as BPA and ILPD. However, it is not entirely clear whether improved targeted ASR arises from:\
a. the transform-triggered formulation itself,\
b. the multi-target loss construction,\
c. or simply differences in attack objective design.\
It would be better to have ablation separating these effects so the improvements can be attributed to the proposed idea. Also, it is not clear about how ASR is defined for the untargeted case.

---

> ### Author Response · Authors · 2026-04-28
> **Response to Reviewer aamU (part 1)**
>
> We thank Reviewer aamU for the constructive review and for acknowledging the interest of our contribution. We address each raised concern below, with corresponding updates reflected in the revised manuscript.
>
> ### (1) Novelty positioning
> We agree with the reviewer's characterization that our optimization (Eq. (3)) involves known tools: PGD, EOT augmentation. Our contribution is not a new optimizer but the introduction and systematic characterization of a previously unrecognized phenomenon: a single imperceptible perturbation can embed multiple distinct malicious behaviors, each conditionally activated by different transform parameters. The adapted tools are the vehicle for studying this phenomenon; the finding that adversarial perturbation has such dynamical transform-dependent properties across architectures, tasks, and image transforms, is our core focus.
>
> ### (2) ASR evaluation and sampling density
> We clarify that ASR is not computed at a small number of 3 isolated points. As stated in Sec. 4.1 Evaluation metrics, for each target parameter $\bar{\theta}_i$i, we sample uniformly over the neighborhood $\mathcal{N}_r(\bar{\theta}_i)$ with interval $0.1$ (or $1$ for JPEG $Q$), yielding multiple evaluation points per range (e.g., $\{0.4, 0.5, 0.6\}$ for $S=0.5$ in scaling). The reported ASR is the average over all samples within each neighborhood. The loss landscape in Fig. 2 further shows that attack effectiveness extends continuously beyond the optimized ranges.
>
> ### (3) Threat model assumptions
> The whitebox setting (known-model and transforms) is standard for studying adversarial phenomena in controlled conditions (as in EOT, PGD, C&W). Since our goal is to characterize a new transform-dependent, multi-target adversarial example, which requires whitebox conditions to begin with, yet we also later on provide blackbox transfer tests in Sec. A.1. Regarding the question on “whether the attacks remain effective under less idealized assumptions”, we have two distinct pieces of evidence in manuscript: (i) Fig. 2 confirms that attacks remain effective when parameters deviate (mismatch) from the optimized values. (ii) Sec. 4.4 camera experiments show that our formation remains effective under variations included by hardwares and physical environment.
>
> ### (4) Discussion of capacity and embedding limits.
> We agree that analysis regarding capacity limits can improve the manuscript further. We provide empirical answers at Fig. 6, stress-tests capacity up to $N=25$ targets and shows that average ASR degrades, with the rate depending on the transform (scaling and JPEG sustain more targets than blur and gamma) and the architecture (ResNet50 accommodates more pairs than ViT-L-16). The loss landscape in Fig. 2 offers geometric intuition that transforms with smoother loss surfaces (blur, gamma) yield wider effective trigger zones, while sharper landscapes (scaling, JPEG) allow finer partitioning.
> The capacity of single perturbation should also directly depend on $\varepsilon$ that larger $\varepsilon$ offer larger learnable parameter ranges with visibility trade-off.
> To summarize, we provided distinct empirical answers for the capacity of perturbation regarding: $N$, transform smoothness, model architectures, yet we agree that a formal capacity analysis (e.g., ASR bounded with involve factors) is a valuable theoretical question that extends beyond provided empirical study. We have updated the manuscript to discuss in detailed at Sec. 5.1.
>
> ### (5) Regarding Definition 1 and benign-set behavior.
> The safe zone is explicitly included in the optimization as one of the $N$ parameter-target pairs with assigning $y_i^* = y$ (groundtruth label). Tab. 2 shows that in the object detection task, ASR at the safe-zone parameters (e.g., $S \sim [0.9, 1.1]$, $\gamma \sim [0.9, 1.1]$) achieves near 100% that model gives groundtruth label at safe zone with the optimized $\delta$, confirming that Eq. (3) objective maintains benign behavior. We hypothesize that this safe zone maintenance is due to targeting benign labels $y$ is consistent with model learned representation.
>
> ### (6) Table 4 clarification (PGD column).
> We apologize for not clearly specifying the PGD setup in Tab. 4, and have revised Sec. 4.4 Image classification to clarify the PGD setup and the intended interpretation of this baseline. The PGD column reports a standard single-target attack (one fixed target, no transform conditioning) as an upper-bound reference for what is achievable in our hardware pipeline. The comparison is not meant to show that our method "outperforms" PGD. Instead, it shows the difficulty of our setting: hitting 91-98\% ASR with two transform-conditioned targets embedded at the same time in a physical setting is close to the upper bound set by an unconstrained single-target attack.

---

> ### Author Response · Authors · 2026-04-28
> **Response to Reviewer aamU (part 2)**
>
> ### (7) Blackbox transfer setup clarification and ablation
>
> We would like to be direct that the blackbox transfer is not our central claim, and Tab. 5 serves as a supplemental transfer test rather than demonstrating “an advanced transfer attack”.
>
> **Untargeted ASR definition.** For the untargeted setting in Tab. 5, an attack is successful if the blackbox model prediction deviates from the ground-truth label $y$ under the queried transform parameter. We have updated Sec. A1 to explicitly state our ASR criterion. When optimize adversarial examples, we use $y$ as target that set $y_i^*$ to $y$ for all $\theta_i$, and solve Eq. (3) (which in this case, collapses to the EOT objective). We then pass generated adversarial examples on ResNet50 to other blackbox models to evaluate the untargeted ASR.
>
> **Ablation: disentangling transform vs. multi-target on blackbox transfer.** We value this suggestion and have included the ablation in Sec. A.2 of the revised manuscript.
>
> We clarify that disentangling these factors under the targeted attack setup is non-trivial: (1) targeted evaluation (i.e., $f(x+\delta)=y$) is inherently asymmetric between multi-target and single-target formulations, and (2) without transforms, multi-target attacks cannot be realized by simply adding an optimized perturbation to the image.
>
> We therefore adopt the untargeted evaluation criterion from Sec.A.1 and design three optimization objectives: (1) transform + multi-target, corresponding to Eq.(3); (2) transform + single-target (EOT), consistent with the untargeted setup in Tab.5; and (3) multi-target without transform. ASRs reported in Tab.~R1 are averaged over transform ranges.
>
> Table R1. Transfer untargeted ASR under objectives disentangling transform and multi-target effects.
> | Trigger | Obj. | Transform | Multi-target | Surrogate ResNet50 | VGG19 | Dense121 | Incv3 | Mobv2 |
> |---------|------|-----------|--------------|--------------------|-------|----------|-------|-------|
> | Scale   | 1    | ✓         | ✓            | 96.50              | 69.90 | 69.90    | 88.00 | 81.70 |
> | Scale   | 2    | ✓         | ✗            | 95.40              | **84.00** | **80.83** | **88.10** | **84.30** |
> | Scale   | 3    | ✗         | ✓            | 96.70              | 63.40 | 64.70    | 86.80 | 75.60 |
> | | | | | | | | | |
> | Blur    | 1    | ✓         | ✓            | 99.50              | 64.50 | 72.90    | 72.80 | 74.20 |
> | Blur    | 2    | ✓         | ✗            | 99.70              | **78.87** | **79.83** | **67.80** | **78.10** |
> | Blur    | 3    | ✗         | ✓            | 99.70              | 64.20 | 66.80    | 69.60 | 70.20 |
>
>
> The ablation reveals that the **transform function is the primary driver of transferability:** Obj. 3 (multi-target without transforms) consistently yields the lowest blackbox ASR, while Obj. 2 (transform with a single target) performs comparably to or above Obj. 1. This is consistent with prior findings that input diversity enhances transfer (DIM, SIM, Admix), confirming that the transferability gains in Tab. 5 stem from transform-based input augmentation rather than the multi-target objective.
> Importantly, this result does not diminish our contribution, as transferability and transform-triggered multi-target behavior are distinct properties. Obj. 2 achieves better transfer because it concentrates the entire perturbation budget on a single target, whereas Obj. 1 must partition that budget across $N$ distinct targets — a strictly harder optimization. We recall that our contribution is not in maximizing blackbox transfer but in identifying and characterizing a new whitebox phenomenon: embedding multiple conditionally-activated behaviors within one perturbation, which Obj. 2 cannot achieve by design. That Obj. 1 retains competitive transfer despite this added constraint further supports the robustness of the transform-dependent phenomenon.
>
> **We have included this ablation with more details in the revision Sec. A.2.**

---

### Review · Reviewer_L1Cp · 2026-04-24

**Summary Of Contributions:**

The paper proposed transform-triggered adversarial examples, which take image transformation as the trigger condition, so that the same adversarial perturbation induces different target behaviors under different transformation parameters, while maintaining normal prediction within the preset safety range.  The experiments verified the effectiveness of this method under transformations such as scaling, blurring, gamma correction and JPEG compression in various classification, object detection models and camera-in-the-loop physical environments.
Key strengths:
1. The paper proposed a novel attack setting which using image transformation as a conditional trigger to make a single disturbance generate multiple attack behaviors.
2. The method can cover multiple transformation types like zoom-in, blurring and can be extended to two types of visual tasks: classification and object detection.
3. The experiments cover not only various models, also certain physical world verifications to a camera-in-the-loop setting.

Key weaknesses:
1. The paper relies on a strong assumption, which the semantics and model predictions should remain largely unchanged after applying a given transformation to a clean image. However, when the transformation is strong, e.g., significant scaling and strong blurring, the assumption can't always hold true.  Then definition is not true f(τ(x;θ))≈y.
2. The physical setup parameters could be more explicit for full reproducibility
3. ASR/ACC can demonstrate the effective of the attack, no metrics to quantify the quality of the boundary between the safe zone and the trigger zone.

**Audience:**

Yes

**Audience Explanation:**

The paper proposes a relatively novel perspective on adversarial attacks, that is, using image transformation as a trigger condition to achieve multi-behavior attacks, which is closely related to research directions such as machine learning security, robustness, and visual model analysis. Although the influence of this work may mainly focus on combating machine learning and related communities, its problem setting and experimental phenomena still have certain research value.

**Claims And Evidence:**

Yes

**Claims Explanation:**

The core claims of the paper have basically been supported by accurate, clear and relatively convincing experiments. The author demonstrated the effectiveness of transform-triggered attacks on multiple models, tasks, and image transformations, and further enhanced the credibility of the results through physical experiments.

However, these evidences mainly prove that the phenomenon exists and is achievable under the current setting, but there is still room for further improvement in terms of its universality, mechanism explanation, and stability in more complex real-world scenarios. In particular, the method relies on a strong premise that the selected transformation should basically maintain the semantics and predictions unchanged on clean images.

**Requested Changes:**

1. The method is based on a strong assumption that the image transformation under consideration should basically maintain the semantics and model predictions unchanged on clean samples. It is suggested that the author discuss more clearly in the main text the role, application scope and possible failure situations of this premise, and explain how the transform parameter ranges were selected in the current experiment to meet this condition
2. The main text mainly adopts a relatively simplified style n=3 transform-target pairs settings. It's better to conduct a more systematic analysis.
3. While the digital attack methodology is largely clear, the physical attack setup in Section 4.4 lacks some specific details for full replication. For instance, precise camera settings (e.g., aperture, ISO, shutter speed) and display model are not provided.
4. Strengthen the distinction between transform-triggered and the existing transform-aware/EOT/transform-based attack
5. While ASR and ACC are appropriate metrics, the definition of uncertainty (e.g., what the shaded regions in Figure 2 represent) is not explicitly stated.
6. A brief analysis of the trade-off between computational cost and attack robustness/effectiveness due to EOT sampling.

---

> ### Author Response · Authors · 2026-04-28
> **Response to Reviewer L1Cp**
>
> We thank Reviewer L1Cp for the positive assessment of our novelty, experimental coverage, and physical validation. We address each requested change below, with corresponding revisions reflected in the updated manuscript.
>
> ### (1) Assumption Eq. (1) and its scope.
>
> Our assumption in Eq. (1) that model predictions on clean examples are preserved under the selected transforms, is a precondition for meaningful evaluation, not a claim that all transforms preserve semantics. Its role is to ensure the fairness that observed misprediction is attributable to the adversarial perturbation rather than transform-induced degradation. Tab. 11 quantifies this directly: clean accuracy remains well above chance for all selected ranges and models (with InceptionV3 at $S=0.5$ being the most sensitive at 29.3\%, which we mitigate by evaluating across a diverse set of eight models). The transform parameters in Sec. 4.2 were selected precisely to stay within the regime where this assumption holds. When a transform is too extreme (e.g., think of a stop sign zoom for $S > 3\times$ might not be presented meaningfully even to humans), clean accuracy itself collapses and the assumption breaks, then our formulation would not apply in that regime. We had added an explicit discussion of Eq.(1) scope in the revision.
>
> ### (2) Scaling beyond $N=3$.
>
> Fig. 6 in Sec. 5.1 (Sec. A.2 in the original manuscript) provides an analysis on optimizing up to $N=25$ targets across four transforms and three architectures. The results show ASR degradation whose rate depends on transform smoothness (scaling and JPEG sustain more targets than blur and gamma) and model architecture (ResNet50 accommodates more pairs than ViT-L-16). We have moved this analysis to main text and discussed it more prominently.
>
> ### (3) Physical setup details.
>
> We included the details to the revised Sec. 4.4: camera model (Nikon Z30), lens (NIKKOR Z DX 16-50mm f/3.5-6.3 VR), display (BenQ BL270 27” 1080p (1920x1080)), capture settings (manual, aperture f/5.6, ISO 100, shutter 1/60s), and working distance (~60 cm). The homography is estimated from four corner markers, and color correction from 1000 randomly sampled RGB pixel pairs, as described in Sec. 4.4.
>
> ### (4) Distinction from EOT and transform-based attacks.
>
> We summarize the key distinction: EOT and transform-based attacks treat transforms as nuisance variables to be marginalized — the goal is a single robust or transferable attack outcome regardless of which transform is applied. Our formulation treats transforms as control variables (trigger): different transform parameters activate different pre-specified target labels from a single perturbation. Tab. 1 in the paper summarizes this contrast. We have strengthened this discussion in Sec. 1 and Tab. 1.
>
> ### (5) Shaded regions in Fig. 2.
>
> The shaded regions represent the standard deviation of the adversarial loss computed over the 1000 images at each transform parameter value. The three colors correspond to three distinct target labels. We have revised the caption to explain it.
>
> ### (6) Computational cost vs. EOT sampling.
>
> Sec. D reports generation times on a single RTX 2080Ti: {2.61, 4.29, 8.67} sec/image for $N = \{3, 5, 10\}$ targets respectively. The cost scales roughly linearly with $N$ since each iteration computes one forward-backward pass per transform-target pair. Within each pair, EOT sampling (drawing multiple $\theta$ from $\mathcal{N}_r(\bar{\theta}_i)$) adds a constant factor per pair. In practice, we found that modest neighborhood sizes ($r = 0.1$) with a few samples suffice for robust trigger zones (as confirmed by the smooth loss valleys in Fig. 2), so the EOT overhead is small relative to the multi-target cost. We have refined Sec. D to discuss computational cost in the revision.

---

### Author Response · Authors · 2026-04-28
**Summary of manuscript changes at mid-discussion phase**

We thank the reviewers for their feedback, and we have incorporated suggestions raised during the discussion phase, in the revised manuscript.

In the submitted revision, the added text is shown in **blue** and removed text is shown in **gray strikethrough**.

We welcome all reviewers to check our updated manuscript and continue the discussion.


Below we summarize the key changes from the initially submitted manuscript:

### **(1) Improved presentation**
1. Sharpened the contribution and its distinction from existing work (Sec. 1).
   - emphasize that our contribution is to characterize an overlooked phenomenon: a single perturbation can induce multiple target behaviors under different transforms. This phenomenon is real, practically relevant, previously overlooked, and worth documenting.
   - acknowledged relevant work TPatch [Zhu et al., USENIX '23], and discussed distinction
2. Provided step-by-step optimization procedures of transform-triggered adversarial examples in both digital and physical setup (Alg. 1,2)
3. Clarified the the shaded regions in Fig. 2 (now noted in the caption).
4. Made the physical setup more specific (Sec. 4.4):
    - added hardware details: display, camera, lens model names and specs, capture settings, and working distance;
    - explain the PGD reference column in Tab. 4 as a setup baseline on the built physical pipeline.


### **(2) Discuss capacity of transform-triggered adversarial examples** (Sec. 5.1)
Promoted the original capacity discussion from the appendix into the main paper to make it more prominent, moving the $N$-up-to-25 stress-test experiment into a new discussion section with formal constraints (Eqs. 6-7).



### **(3) Blackbox transfer ablation** (Sec. A.2 / Tab. 7)
Added an ablation that disentangles the transform function from the multi-target objective, showing that the transform function is the primary driver of transferability — consistent with prior input-diversity findings and orthogonal to our whitebox multi-behavior contribution.

---

> ### Author Response · Authors · 2026-05-12
> **request any additional comments**
>
> Dear Reviewers and AE,
>
> We hope you had a chance to read our responses and updated manuscript. We also hope that our explanations and experiments addressed your concerns. Please let us know if you have any additional comments or need any additional information or clarification from us.
>
> Thanks!

---

### Decision · Action_Editor_dovo · 2026-05-29

**Recommendation:** Accept with minor revision

**Additional Comments:**

The recommendation is based on the reviewers' comments, the action editor's evaluation, and the authors’ response.

This paper studies the effect of image transformations on adversarial examples for images. All reviewers agree that the empirical results support the findings. However,  although the authors’ rebuttal has successfully addressed the major concerns of reviewers, the reviewers also highlight the lack of sufficient technical novelty compared to existing works. Nonetheless, based on TMLR's review criterion (https://jmlr.org/tmlr/acceptance-criteria.html), the novelty concern has been dismissed.

Overall, I recommend acceptance of this submission with minor revision. In the final version, please polish Section 3.3 (Physical Attacks with Hardware-in-the-Loop) and discuss novel insights to the considered physical attacks or defenses setups. I also expect the authors to include the new results and suggested changes during the rebuttal phase to the final version.

**Audience:**

Yes

**Audience Explanation:**

Of broad interest to adversarial robustness, AI security, and safety.

**Claims And Evidence:**

Yes

**Claims Explanation:**

The claims of transform-triggered adversarial examples are supported by the empirical results.